# Reconfigurable self-assembly of photocatalytic magnetic microrobots for water purification

Mario Urso ®[1], Martina Ussia ®[1], Xia Peng[1], Cagatay M. Oral[1] & Martin Pumera ®[1,2,3,4] ✉

The development of artificial small-scale robotic swarms with nature-mimicking collective behaviors represents the frontier of research in robotics. While microrobot swarming under magnetic manipulation has been extensively explored, light-induced self-organization of micro- and nanorobots is still challenging. This study demonstrates the interaction-controlled, reconfigurable, reversible, and active self-assembly of $TiO_2/\alpha\text{-}Fe_2O_3$ microrobots, consisting of peanut-shaped $\alpha\text{-}Fe_2O_3$ (hematite) microparticles synthesized by a hydrothermal method and covered with a thin layer of $TiO_2$ by atomic layer deposition (ALD). Due to their photocatalytic and ferromagnetic properties, microrobots autonomously move in water under light irradiation, while a magnetic field precisely controls their direction. In the presence of $H_2O_2$ fuel, concentration gradients around the illuminated microrobots result in mutual attraction by phoretic interactions, inducing their spontaneous organization into self-propelled clusters. In the dark, clusters reversibly reconfigure into microchains where microrobots are aligned due to magnetic dipole-dipole interactions. Microrobots' active motion and photocatalytic properties were investigated for water remediation from pesticides, obtaining the rapid degradation of the extensively used, persistent, and hazardous herbicide 2,4-Dichlorophenoxyacetic acid (2,4D). This study potentially impacts the realization of future intelligent adaptive metamachines and the application of light-powered self-propelled micro- and nanomotors toward the degradation of persistent organic pollutants (POPs) or micro- and nanoplastics.

Nature provides a plethora of self-organization phenomena, where a physical or biological system manifests a global pattern resulting from the interactions between its constituents: in deserts, sand grains form beautiful rippled dunes under the action of wind, while in living beings, cells structure into tissues to perform precise functions[1]. In this context, some animal systems can organize into diverse patterns to accomplish different tasks. An example of reconfigurable self-organization is represented by fire ants, which build living bridges to march across gaps or assemble into floating rafts to survive floods[2,3]. Necessity-driven collective behaviors like these have fascinated

[1]Future Energy and Innovation Laboratory, Central European Institute of Technology, Brno University of Technology, Purkyňova 123, 61200 Brno, Czech Republic. [2]Advanced Nanorobots & Multiscale Robotics Laboratory, Faculty of Electrical Engineering and Computer Science, VSB—Technical University of Ostrava, 17. listopadu 2172/15, 70800 Ostrava, Czech Republic. [3]Department of Medical Research, China Medical University Hospital, China Medical University, Hsueh-Shih Road 91, 40402 Taichung, Taiwan. [4]Department of Chemical and Biomolecular Engineering, Yonsei University, Yonsei–ro 50, Seodaemun–gu, 03722 Seoul, Republic of Korea. ✉e-mail: pumera.research@gmail.com

researchers in the robotics field, who learned from nature and aimed to mimic it by developing micro- and nanorobot swarms to solve specific challenges beyond individuals' capabilities[4–9].

Micro- and nanorobots represent the latest development of micro- and nanoscale materials, obtained by introducing the motion feature and increasing intelligence in terms of programmable functions and behaviors[10–12]. These small-scale devices are powered by local chemical fuels (e.g., $H_2O_2$), energy fields (light, magnetic fields), or a self-motile biological component (algae, sperm) and applied to improve the performance of non-motile objects in water remediation, sensing, or medical therapies[13–19]. Among the different power sources for micro- and nanorobots, light is particularly advantageous, being a natural, abundant, and renewable form of energy[20]. The light-driven self-propulsion relies on breaking the symmetry of a photoactive material (e.g., a photocatalytic semiconductor) so that, under light irradiation, it produces an asymmetric gradient of solute concentration (self-diffusiophoresis), electric potential (self-electrophoresis) or temperature (self-thermophoresis), inducing its movement[21]. An effective strategy to break a semiconductor microparticles' symmetry is depositing a noble metal layer. For instance, $TiO_2$, one of the most used photocatalysts due to its high photocatalytic efficiency, stability, and safety, has been combined with different noble metals to produce efficient UV light-powered micro- and nanorobots[22–25]. Alternatively, $TiO_2$ has been deposited onto passive particles to make them move under UV light irradiation[26]. Heterostructures between distinct semiconductors represent a promising solution to noble metals' high cost and potentially hazardous corrosion[27,28]. In this context, $\alpha$-$Fe_2O_3$ represents an advantageous building block due to its peculiar properties, such as biocompatibility, photoactivity under visible light, magnetism, and cheap preparation methods[29]. Thus, it can be used to devise microrobots with both photocatalytic and magnetic properties, which offer several benefits for various applications. They can utilize light to move and degrade water pollutants, while magnetic fields allow easy collection after the treatment. If the medium's properties hinder the light-driven self-propulsion, magnetic fields can provide powerful movement, and the light source can activate their photocatalytic activity. Furthermore, magnetic fields enable precise navigation of microrobots into hard-to-reach areas, such as inside pipelines in water remediation applications or specific regions of the human body in biomedical applications.

Highly controlled reconfigurable self-organization in robotic systems has been mainly obtained using magnetic fields, which allowed for programming reversible swarming of $\alpha$-$Fe_2O_3$ microrobots into liquid, chain, vortex, and ribbon-like patterns or magnetic navigation of microrobot swarms with adaptivity to environmental changes through a deep learning-based real-time planning strategy[30,31]. Magnetic fields offer several adjustable parameters that govern the assembly of magnetic micro- and nanorobots. However, their implementation necessitates relatively bulky and expensive magnetic setups. Achieving such a degree of control solely using light sources is desirable yet challenging. Nevertheless, $\alpha$-$Fe_2O_3$/polysiloxane hybrid colloids organized into reconfigurable structures under the action of UV light or magnetic fields in 1-3% $H_2O_2$[32]. Interestingly, when enriched with hydroxyl groups ($OH^-$), $TiO_2$ micromotors gathered into flocks, which dilated upon UV light irradiation in 0.5% $H_2O_2$ leading to micromotors' collective motion[33]. Also, these micromotors arranged themselves into elongated shapes when transiting a narrow microfluidic channel. Colloidal surfers based on polymer/$\alpha$-$Fe_2O_3$ microparticles self-organized into two-dimensional (2D) living crystals when exposed to light in 0.1–3% $H_2O_2$ at basic pH conditions[34]. Whereas, under UV light irradiation in 1.5% $H_2O_2$ and neutral pH conditions, active $TiO_2$/$SiO_2$ Janus particles attracted passive colloids, functioning as nucleation centers for the growth of 2D crystals whose size and symmetry were controlled by light intensity and size ratio between active and passive particles[35]. Another approach exploited the incident

light angle for programming the self-assembly of $Pt$/$TiO_2$ micromotors into active or static planar crystals and phototactic micromotor streams[36]. The formation of these superstructures relies on the interplay between light-induced self-propulsion and attractive osmotic or phoretic interactions. Therefore, by turning off the light source, the self-assembled configurations are broken. Besides, using a combination of magnetic and photocatalytic materials, micromotors were arranged into chain-like structures through the application of a magnetic field, while their movement in water or $H_2O_2$ was facilitated by light irradiation[37,38]. In this manner, the assembly or disassembly of microchains is achieved by activating or deactivating the magnetic field. Instead, thanks to magnetic dipole–dipole interactions and a peculiar shape, spontaneous chain formation was achieved for $Pt$/$\alpha$-$Fe_2O_3$ cubic microrobots without external stimuli[39]. Still, for spontaneously formed microchains, there was no control over their assembly/disassembly, unlike those manipulated by magnetic fields, or reconfigurability. In all cases, the properties of the self-organized state, such as reconfigurability, reversibility, motility, and stability, are dictated by how constituent microrobots interact. Therefore, a profound understanding of these interactions would promote the development of mobile adaptive robotic swarms.

Reconfigurable self-assembly of microrobots has significant potential for various applications, including water purification. By responding to external stimuli such as light or magnetic fields, microrobots can be programmed to assemble into porous networks or complex aggregates that trap water pollutants and degrade them through photocatalytic reactions. In addition, microrobots can surround solid impurities, such as micro- and nanoplastics, and break them down when activated by light. Alternatively, microrobots can reconfigure into linear structures and use magnetic fields to mop up plastic debris from water sources. Moreover, microrobots can selectively detect and bind to these particles in water, providing a visual indication of their presence and distribution. Finally, microrobots' magnetic properties facilitate their collection and removal from the treated water.

Inspired by fire ants' self-organization behavior into linear (bridges) or planar (rafts) structures, this work demonstrates the interaction-controlled, reconfigurable, reversible, and active self-assembly of light-powered magnetic $TiO_2$/$\alpha$-$Fe_2O_3$ microrobots into clusters and microchains under optical stimuli, without using magnetic fields (Fig. 1). Noble metal-free microrobots were prepared by $TiO_2$ ALD on peanut-shaped $\alpha$-$Fe_2O_3$ microparticles synthesized by a hydrothermal method, creating an environmentally friendly heterojunction. The microrobots showed self-propulsion under UV light irradiation in $H_2O_2$-free water and precise navigation along a predetermined path using an external magnetic field to steer their motion. In a low-concentrated $H_2O_2$ solution, they spontaneously assembled into self-motile clusters, resembling floating rafts of fire ants, due to $H_2O_2$ gradient-induced attractive phoretic interactions under UV light irradiation, as supported by numerical simulations. By switching off UV light, clusters reconfigured into microchains, emulating bridges of fire ants, thanks to magnetic dipole–dipole interactions between microrobots. Multiple on/off switching of UV light irradiation proved the reversibility of the self-assembly process. Furthermore, because of their high photocatalytic activity, microrobots were applied in the remediation of polluted water to accelerate the degradation of the herbicide 2,4D via the synergistic combination of self-propulsion and photocatalysis.

## Results and discussion
### Fabrication and characterization of $TiO_2$/$\alpha$-$Fe_2O_3$ microrobots
Figure 2a illustrates the fabrication process for $TiO_2$/$\alpha$-$Fe_2O_3$ microrobots. $\alpha$-$Fe_2O_3$ microparticles were synthesized by a facile hydrothermal reaction and dispersed onto glass slides to deposit a thin $TiO_2$ layer (-30 nm) on their surface by ALD. The scanning electron

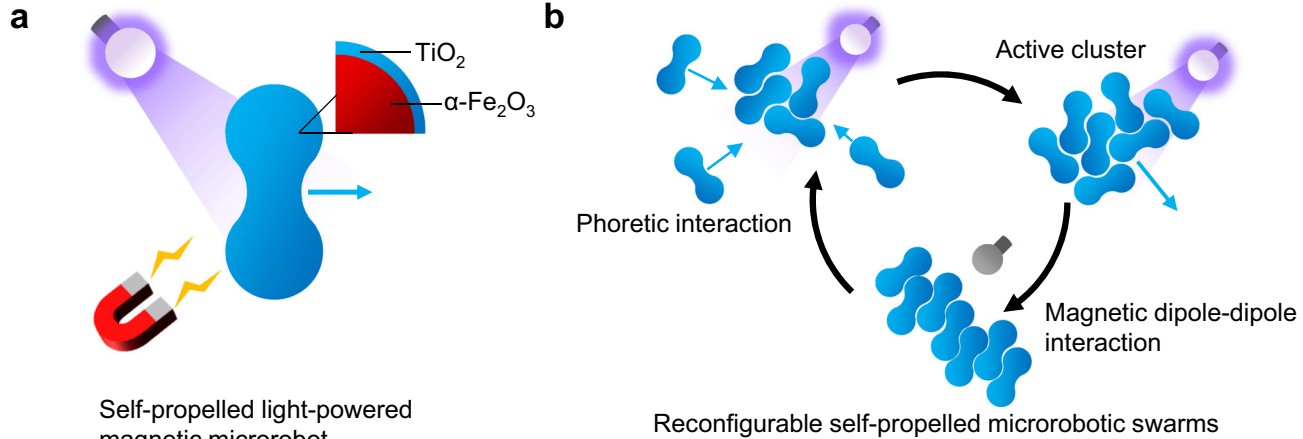

**Fig. 1 | Reconfigurable self-assembly of TiO₂/α-Fe₂O₃ microrobots. a** Self-propelled, light-powered, magnetically navigable, peanut-shaped TiO₂/α-Fe₂O₃ microrobot. **b** Fire ants-mimicking, reconfigurable, reversible, and active self-assembly of TiO₂/α-Fe₂O₃ microrobots into planar (cluster) and linear (microchain) structures mediated by attractive phoretic interaction or magnetic dipole–dipole interaction under optical stimuli.

microscopy (SEM) image in Fig. 2b reveals the peanut-like shape of α-Fe₂O₃ microparticles. Their dimensions were accurately measured using several scanning transmission electron microscopy (STEM) images, such as Fig. 2c, which indicates their high uniformity. By analyzing more than 60 microparticles, it was estimated that their long axis ($a$) measures $2.44 \pm 0.09\,\mu m$, while their short axis ($b$) measures $1.12 \pm 0.05\,\mu m$. SEM and STEM analyses of TiO₂/α-Fe₂O₃ microrobots indicated that TiO₂ deposition did not affect their shape and size (Supplementary Fig. 1). Although the TiO₂ layer could not be distinguished by SEM, energy dispersive X-ray (EDX) spectroscopy provided the first proof of successful TiO₂ deposition. In fact, the elemental mapping images of a group of microrobots in Fig. 2d show the uniform distribution of O, Fe, and Ti within the structure. It is worth noting that they do not present the characteristic Janus structure behind the motion ability of several metal/semiconductor micro- and nanorobots, including α-Fe₂O₃ microspheres[40]. Instead, TiO₂ growth by ALD resulted in a conformal coating of α-Fe₂O₃ microparticles. This observation agrees with a previous study reporting the full coverage of Mg microspheres with ALD-deposited TiO₂ except for the region in contact with the substrate[41]. Although such uncovered areas were not directly observed in the SEM images of TiO₂/α-Fe₂O₃ microrobots, their presence is expected because of the physical contact between α-Fe₂O₃ microparticles and the substrate, leading to a shadowing effect during ALD deposition of TiO₂.

The crystalline structure of TiO₂/α-Fe₂O₃ microrobots was investigated by X-ray diffraction (XRD). The measurement was operated in grazing incidence mode to reduce X-ray penetration and enhance surface sensitivity. The acquired XRD pattern is shown in Fig. 2e, where the characteristic peaks of α-Fe₂O₃ crystal planes were detected (PDF card #00-001-1053)[42]. Precisely, the peaks at $2\theta$ values of 24.16°, 33.28°, 35.74°, 40.99°, 49.50°, 54.23°, 57.56°, 62.26°, 64.18°, and 72.03° correspond to (012), (104), (110), (113), (024), (116), (122), (214), (300), (1010) planes, respectively. Instead, the peak at $2\theta = 25.4°$ was ascribed to (101) surface of anatase TiO₂ (JCPDS card no. 21-1272), which is considered the most photoactive one among TiO₂ phases, holding considerable promise for microrobots light-driven motion and water purification applications[43].

X-ray photoelectron spectroscopy (XPS) allowed the study of the surface chemical states of α-Fe₂O₃ microparticles and TiO₂/α-Fe₂O₃ microrobots. Survey and fitted high-resolution spectra are compared in Supplementary Figs. 2 and 3 (the binding energy values for all fitted peaks are reported in Supplementary Table 1). Figure 2f displays the high-resolution spectrum of Fe 2$p$ for α-Fe₂O₃ microparticles and Ti 2$p$ for TiO₂/α-Fe₂O₃ microrobots. The peaks at 711.2 eV and 724.6 eV

binding energy correspond to Fe 2$p_{3/2}$ and Fe 2$p_{1/2}$, with the related satellite peaks at 719.1 eV and 732.3 eV, attributed to $Fe^{3+}$ at octahedral sites in α-Fe₂O₃[44]. Each peak was fitted by several components, reflecting a complex multiplet splitting in agreement with the literature[45]. Fe 2$p$ signal was almost absent for TiO₂/α-Fe₂O₃ microrobots that exhibited Ti 2$p_{3/2}$ and Ti 2$p_{1/2}$ signals at 458.5 eV and 464.2 eV binding energy, confirming the conformal coating of α-Fe₂O₃ microparticles surface by TiO₂[46]. Three components were identified in the high-resolution spectra of O 1$s$ for both α-Fe₂O₃ microparticles and TiO₂/α-Fe₂O₃ microrobots (Supplementary Fig. 3). The first one was assigned to the oxides (530.0 eV for α-Fe₂O₃ and 529.7 eV for TiO₂), the second one to the corresponding hydroxides (731.1 eV for FeOOH and 731.6 eV for TiOOH), and the third one, at higher binding energies, to adsorbed H₂O molecules[45,46].

**Microrobots' motion and collective behaviors**

First, the motion behavior of α-Fe₂O₃ microparticles and TiO₂/α-Fe₂O₃ microrobots was investigated in pure water. The microparticles displayed no autonomous movement under UV light exposure, while an opposite scenario was found for the microrobots. Supplementary Fig. 4 reports the time-lapse micrographs (extracted from Supplementary Movie 1) of a microrobot's trajectory during four cycles of 5 s on/off switching of UV light and the corresponding instantaneous speed as a function of time. The microrobot shows Brownian motion in the dark condition and self-propulsion on the microscope's focal plane (i.e., on top of the glass slide) under UV light irradiation. Moreover, the microrobot's speed increases when UV light turns on and decreases during the self-propulsion. Therefore, microrobots' speed in the last illumination interval appears significantly lower than the first one. Microrobots were exposed to prolonged UV light irradiation to investigate this intriguing phenomenon further. Figure 3a reports a microrobot's representative trajectory and instantaneous speed (color-coded) under UV light irradiation in pure water, corresponding to Supplementary Movie 2. The speed is initially ~15 μm s⁻¹ and rapidly decreases until reaching a constant value of ~2.5 μm s⁻¹. A similar decay in speed was observed for a photochemically powered AgCl Janus micromotor[47]. In that case, the motion mechanism relied on ionic self-diffusiophoresis due to the release of ions upon the light-induced conversion of AgCl into Ag. Consequently, after some time, the micromotor stopped due to the AgCl consumption and underwent Brownian motion. On the contrary, Figure 3a suggests that TiO₂/α-Fe₂O₃ microrobots continue to propel at a lower, constant speed, from now on referred to as final speed, without manifesting Brownian motion. Fig. 3b compares the decline of the average instantaneous

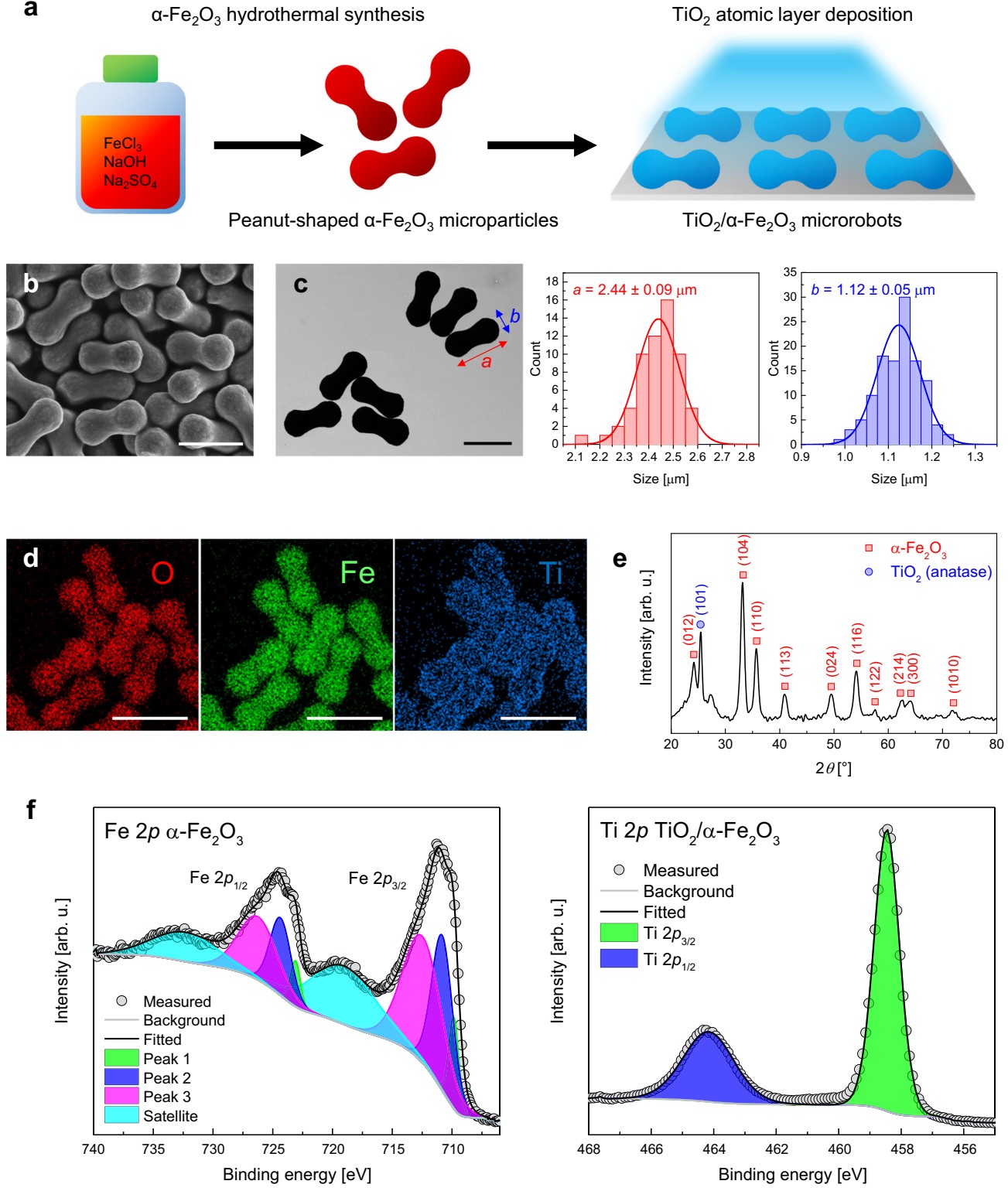

**Fig. 2 | Fabrication and characterization of TiO₂/α-Fe₂O₃ microrobots. a** Scheme of the fabrication steps. **b** SEM and **c** STEM images of α-Fe₂O₃ microparticles and their size distributions obtained from the analysis of STEM images by measuring $n = 60$ independent microparticles, where $a$ represents their long axis and $b$ represents their short axis. Scale bars are $2\,\mu m$. **d** EDX elemental mapping images showing the distribution of O, Fe, and Ti in TiO₂/α-Fe₂O₃ microrobots. Scale bars are $2.5\,\mu m$. **e** XRD pattern of TiO₂/α-Fe₂O₃ microrobots. **f** High-resolution XPS spectra of Fe 2p for α-Fe₂O₃ microparticles and Ti 2p for TiO₂/α-Fe₂O₃ microrobots.

speed as a function of time for 20 microrobots under UV light irradiation in pure water and 1% H₂O₂. In pure water, microrobots' initial speed is ~7.5 µm s⁻¹ and reaches the final speed of ~2.5 µm s⁻¹ within 20 s from the beginning of UV light irradiation. H₂O₂ resulted in a slightly higher initial speed of ~12.5 µm s⁻¹, which dropped to a similar final

speed in less than 7.5 s. The faster decay in 1% H₂O₂ suggests that it enhances the mechanism responsible for the deceleration of the microrobots.

A control experiment demonstrated that α-Fe₂O₃ microparticles under prolonged UV light irradiation in pure water only display the

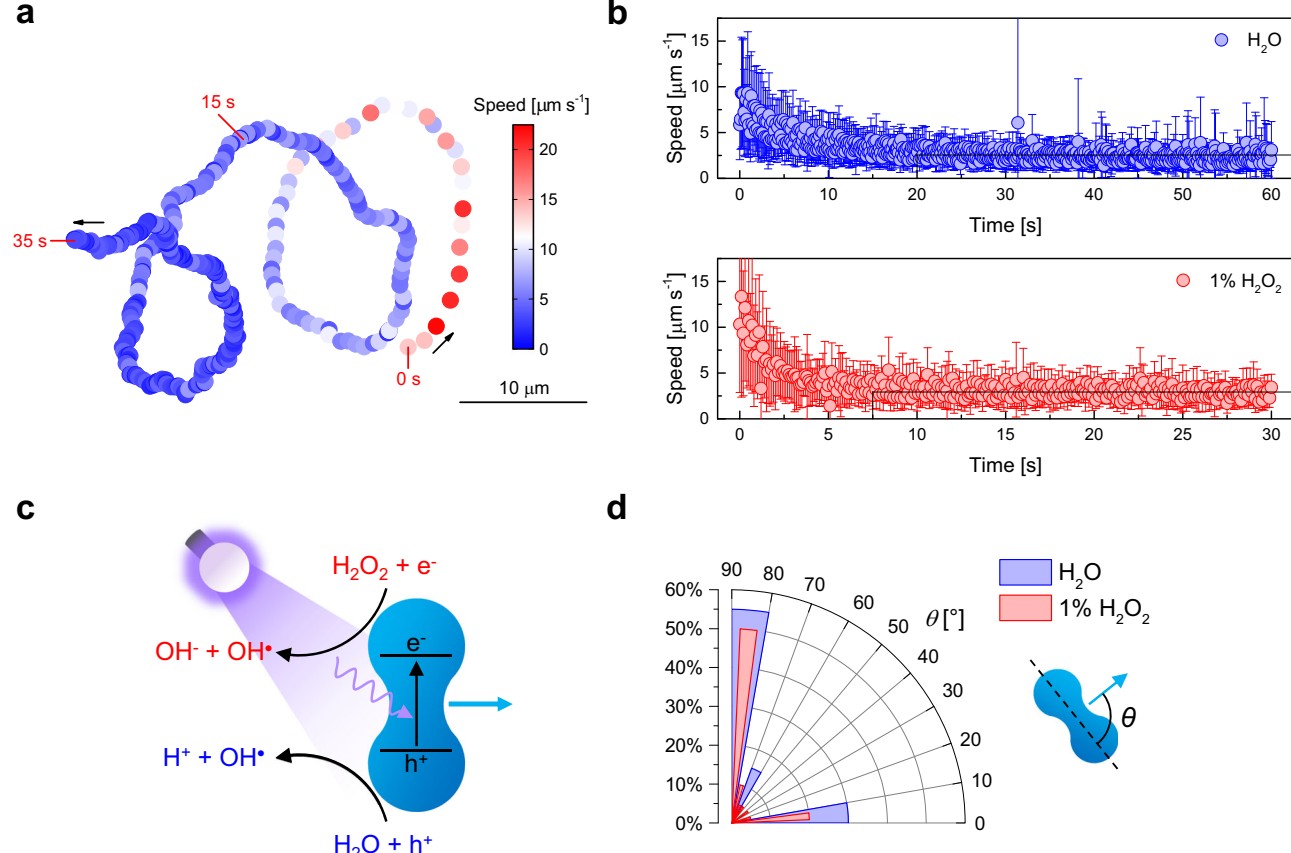

**Fig. 3 | Light-powered motion of TiO$_2$/α-Fe$_2$O$_3$ microrobots. a** Representative trajectory of a TiO$_2$/α-Fe$_2$O$_3$ microrobot with decreasing instantaneous speed (color-coded) under UV light irradiation in pure water for 35 s. **b** Instantaneous speed as a function of time of TiO$_2$/α-Fe$_2$O$_3$ microrobots under UV light irradiation in pure water and 1% H$_2$O$_2$. Error bars represent the standard deviation, $n$ = 20

independent microrobots. **c** Scheme of the light-powered motion mechanism of TiO$_2$/α-Fe$_2$O$_3$ microrobots. **d** Motion direction of $n$ = 20 TiO$_2$/α-Fe$_2$O$_3$ microrobots under UV light irradiation in pure water and 1% H$_2$O$_2$, described by the angle $θ$ to their long axis.

characteristic random trajectories of Brownian motion with a low and constant speed of ~1 μm s$^{-1}$ (Supplementary Fig. 5). Consequently, α-Fe$_2$O$_3$ may not have a major contribution in diminishing microrobots' speed. Instead, similar to the previously mentioned AgCl-based micromotor, a partial consumption of the engine, i.e., the TiO$_2$ coating, was assumed to explain the decrease of the speed of TiO$_2$/α-Fe$_2$O$_3$ microrobots. To verify this hypothesis, an experiment was performed by exposing microrobots to UV light irradiation in 1% H$_2$O$_2$ for 5 min. Microrobots' surface was then characterized by XPS and compared to untreated microrobots in Supplementary Fig. 6. While microrobots originally showed Ti 2$p_{3/2}$ and Ti 2$p_{1/2}$ peaks in the Ti 2$p$ region and no signal in the Fe 2$p$ region, treated microrobots exhibited Fe 2$p_{3/2}$ and Fe 2$p_{1/2}$ peaks together with Ti 2$p$ peaks. This result suggests that the uniform TiO$_2$ layer underwent partial corrosion, exposing a fraction of the surface of α-Fe$_2$O$_3$ microparticles. Therefore, it is reasonably concluded that the initial deceleration is attributed to the degradation of TiO$_2$. Nevertheless, the lifetime of the microrobots was tested under UV light irradiation in pure water for 60 min, and it was found that the microrobots maintained their self-propulsion ability and speed (Supplementary Fig. 7).

In light of this, the movement of TiO$_2$/α-Fe$_2$O$_3$ microrobots is explained according to the scheme in Fig. 3c. Upon UV light irradiation, the TiO$_2$ layer absorbs photons with sufficient energy to excite electrons from the semiconductor's valence band to the conduction band. The photogenerated electron-hole pairs react with surrounding water molecules, producing a charged product concentration gradient responsible for microrobots' autonomous motion via electrolyte

self-diffusiophoresis[48]. A control experiment in a concentrated salt solution (0.1 M NaCl) confirmed the proposed mechanism since the high ionic strength of the media hindered microrobots' motility. The H$_2$O$_2$ fuel, when present, contributes to the propulsion process through supplementary chemical reactions, as shown in Fig. 3c. This leads to increased speed and faster deceleration of microrobots, compared to the case of pure water, during the initial stage of UV light irradiation.

Earlier studies on α-Fe$_2$O$_3$ micromotors required surface activation through the use of concentrated acid solutions, high amounts of H$_2$O$_2$ (ranging from 1% to 10%), and an increase of the medium pH (~8.5) to achieve the self-propulsion capability[49,50]. However, the application of a TiO$_2$ coating circumvents the need for potentially hazardous pre-utilization steps, toxic H$_2$O$_2$, and pH adjustments. Still, those studies reported contradicting results regarding the preferential motion direction, perpendicular or parallel, to the microparticles' long axis. For this reason, the angle $θ$ between the motion direction and the long axis of TiO$_2$/α-Fe$_2$O$_3$ microrobots was measured. Figure 3d shows that, in water, 55% of microrobots preferentially move perpendicularly to the long axis ($θ$ ~90°), while 30% move parallelly to the long axis ($θ$ ~0°). The same trend was observed in H$_2$O$_2$, but the fraction of microrobots moving with 10° < $θ$ < 80° slightly increased.

Microrobots' mobility was also examined under visible light irradiation, using blue light (Supplementary Fig. 8). No self-propulsion was observed in pure water; however, upon the introduction of H$_2$O$_2$, microrobots demonstrated the ability to self-propel. This behavior can be attributed to the absorption of blue light by α-Fe$_2$O$_3$, considering

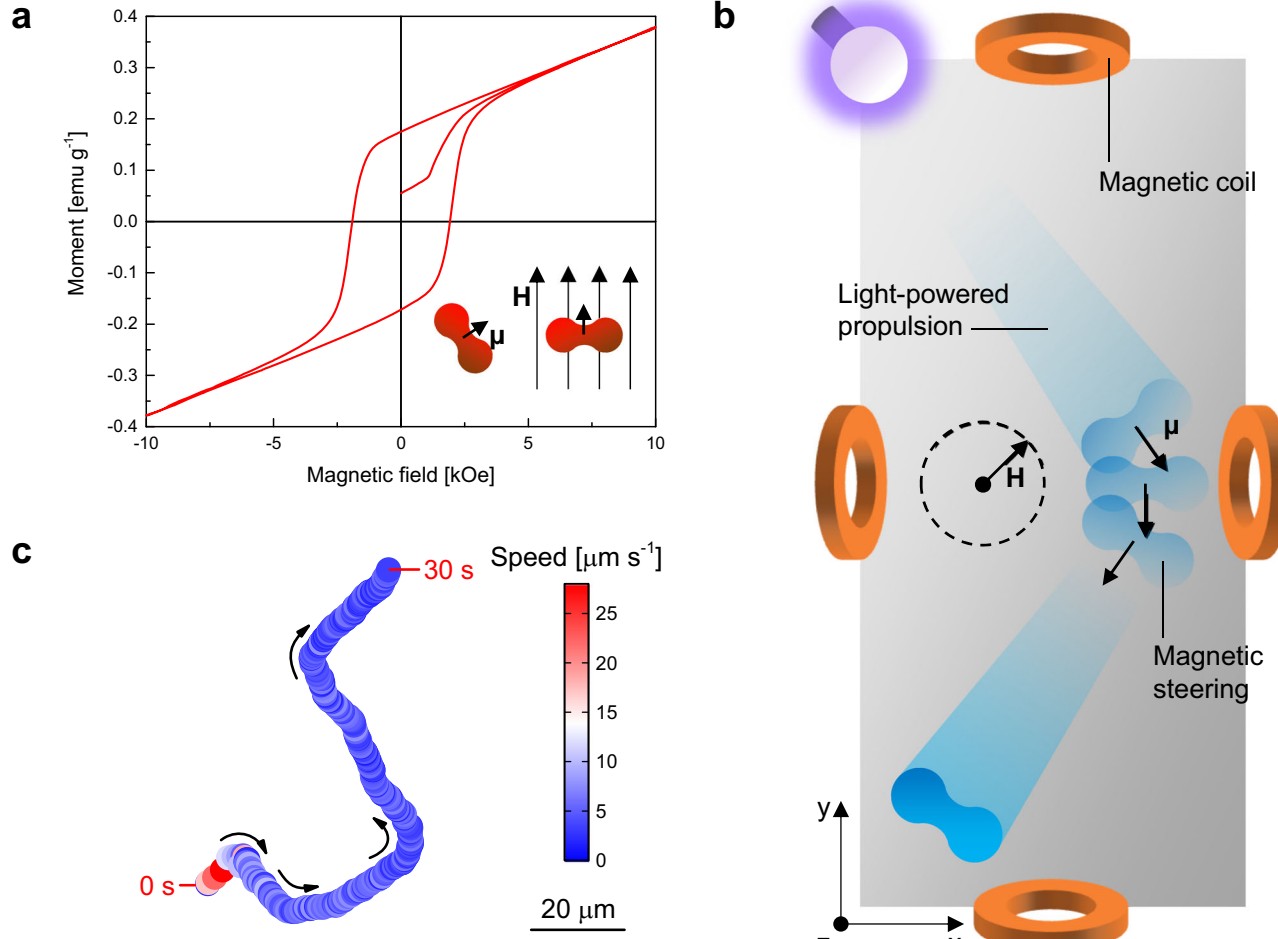

**Fig. 4 | Magnetic navigation of light-powered TiO$_2$/α-Fe$_2$O$_3$ microrobots.**
**a** Magnetic hysteresis loop of α-Fe$_2$O$_3$ microparticles. The inset illustrates the magnetic dipole moment (**μ**) of an α-Fe$_2$O$_3$ microparticle and its orientation according to the direction of an applied magnetic field (**H**). **b** Scheme of the magnetic steering of a TiO$_2$/α-Fe$_2$O$_3$ microrobot under UV light irradiation using the rotation of a magnetic field (**H**) on the xy plane, generated by a magnetic setup consisting of orthogonal coil pairs. **c** Representative trajectory of a TiO$_2$/α-Fe$_2$O$_3$ microrobot with decreasing instantaneous speed (color-coded) under simultaneous application of UV light irradiation and a magnetic field, whose directionality was steered by ~90° several times, in pure water for 30 s.

the larger bandgap of TiO$_2$. Notably, no speed decay was detected in the initial stage of the experiment, unlike for UV light irradiation. Furthermore, microrobots' speed was measured at increasing concentrations of H$_2$O$_2$, finding a minor increase in speed from 0.5 to 2% H$_2$O$_2$.

In addition to the light-driven autonomous locomotion of TiO$_2$/α-Fe$_2$O$_3$ microrobots, precise control over their directionality can be obtained by exploiting α-Fe$_2$O$_3$ microparticles' magnetic properties, which were characterized by a vibrating sample magnetometer (VSM). The measured magnetic hysteresis loop in Fig. 4a is the fingerprint of an antiferromagnetic or weakly ferromagnetic material with 0.37 emu g$^{-1}$ remanence and −1.9 kOe coercivity. Peanut-shaped α-Fe$_2$O$_3$ microparticles possess a magnetic dipole moment perpendicular to their long axis[29]. Thus, when immersed in a magnetic field, they orient so that their magnetic dipole moment parallels the magnetic field, as depicted in Fig. 4a inset. Previous studies have utilized magnetic fields to guide peanut-shaped α-Fe$_2$O$_3$ micromotors for various applications, including the non-contact manipulation of cells[51]. In the case of TiO$_2$/α-Fe$_2$O$_3$ microrobots, their combined light-powered motion and magnetic responsiveness provide additional possibilities for accurate navigation within a liquid medium. For this purpose, a magnetic setup generating a rotating magnetic field on the xy plane was used. As depicted in Fig. 4b, microrobots move due to UV light irradiation while their direction is continuously adjusted by changing the angle of the applied magnetic field. Figure 4c reports the trajectory of a UV light-driven and magnetically steered microrobot, whose movement was turned by ~90° several times by adequately orienting the direction of the magnetic field (Supplementary Movie 3). These results are promising for those applications where a high control over the position of microrobots is compulsory, such as cargo transport[52–54].

TiO$_2$/α-Fe$_2$O$_3$ microrobots manifested a fascinating collective behavior in the presence of H$_2$O$_2$ since light exposure allowed them to rapidly switch from two different self-assembled states: dynamic clusters and static microchains. Under UV light irradiation, microrobots moved in 1% H$_2$O$_2$ until colliding each other, forming active clusters. Supplementary Fig. 9 includes time-lapse micrographs that demonstrate the aggregation of the microrobots and cluster configuration. These clusters grew with time since they continuously attracted and trapped microrobots moving in the surrounding area. Once the UV light was turned off, the clusters quickly fragmented into long chains consisting of microrobots. This behavior is visualized in the time-lapse micrographs in Fig. 5a, extracted from Supplementary Movie 4. After 60 s of UV light irradiation, the microrobots arranged into clusters of random size and shape, leaving a few free microrobots in the field of view. However, almost all clusters disappeared after 60 s in the dark (120 s since the beginning of the experiment), leaving microchains or isolated microrobots. Of note, microrobots reversibly

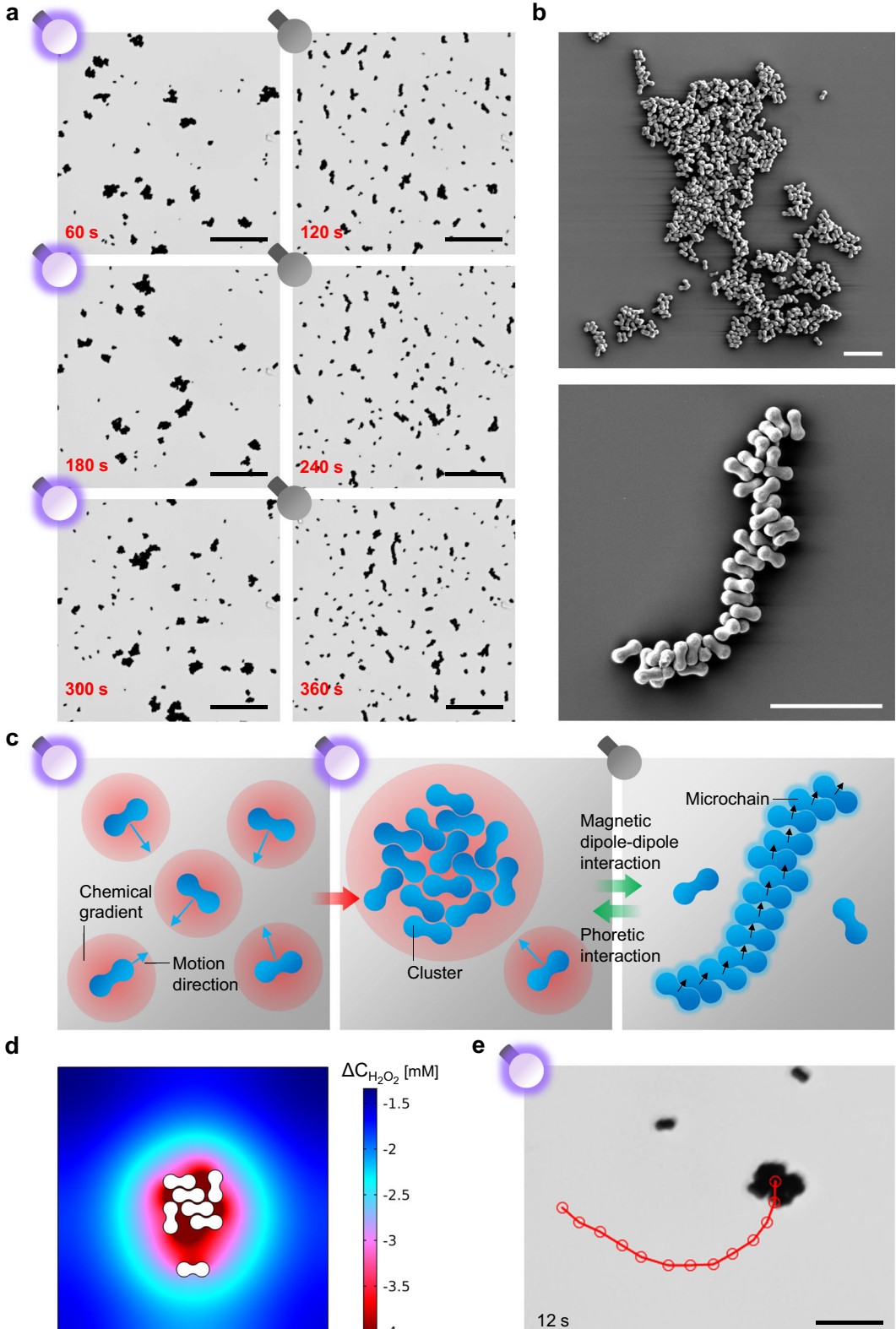

**Fig. 5 | Interaction-controlled, reconfigurable, reversible, and active self-assembly of TiO₂/α-Fe₂O₃ microrobots. a** Time-lapse micrographs at 60 s on/off switching of UV light irradiation, showing three cycles of microrobot clustering under UV light irradiation in 1% $H_2O_2$ and their reconfiguration into microchains in the dark. Scale bars are 50 μm. **b** SEM images of a microrobot cluster and a microchain. Scale bars are 10 μm. **c** Scheme of the reconfigurable and reversible self-assembly process of TiO₂/α-Fe₂O₃ microrobots under UV light irradiation in $H_2O_2$, mediated by phoretic and magnetic dipole–dipole interactions. **d** Simulated $H_2O_2$ concentration gradient around a cluster of TiO₂/α-Fe₂O₃ microrobots under UV light irradiation in 1% $H_2O_2$, resulting in a pressure imbalance inducing the attraction of a nearby TiO₂/α-Fe₂O₃ microrobot. **e** Micrograph showing the representative trajectory of a TiO₂/α-Fe₂O₃ microrobot cluster under UV light irradiation in 1% $H_2O_2$ for 12 s. The scale bar is 10 μm.

switched from one state to another for several cycles (three cycles in this experiment). Furthermore, the state cluster/microchain was preserved as long as the UV light irradiation was maintained on/off status. This feature allowed for freezing the microrobots in the two states and acquiring their SEM images. Particularly, the UV light was kept to force microrobot self-assembly into clusters until the liquid medium evaporated. Then, the SEM image of a cluster was recorded and reported in Fig. 5b (top) revealing hundreds of gathered microrobots. The voids within the microrobot cluster are attributed to the evolution of $O_2$ bubbles under constant UV light exposure, pushing nearby microrobots. Similarly, microchains, formed after clusters' disgregation in the dark, were undisturbed until the liquid medium evaporated. Afterward, the SEM image of the microchain in Fig. 5b (bottom) was recorded, showing tens of adjacent microrobots aligned along their long axis in a snake-like shape.

These reconfigurable microrobot collectives originate from the different interactions between their constituents, as schematically illustrated in Fig. 5c. Under UV light irradiation, the $TiO_2$ layer induces $H_2O_2$ decomposition, producing an $H_2O_2$ gradient around the microrobot. The gradient, in turn, generates an osmotic pressure that attracts nearby microrobots. A bigger cluster produces a larger $H_2O_2$ concentration gradient, as demonstrated by the numerical simulation in Fig. 5d, leading to cluster growth. Therefore, phoretic interactions due to pressure imbalances are at the basis of clusters' formation and growth. This behavior is a signature of self-diffusiophoretic colloidal micromotors[55]. A similar mechanism explained the formation of self-assembled structures of hybrid $Fe_2O_3$/polysiloxane colloids[32]. In contrast, in the absence of UV light, the consumption of $H_2O_2$ and, thus, phoretic interactions cease, releasing the microrobots. Nonetheless, in this stage, the magnetic dipole–dipole interactions between microrobots prevail. Consequently, microrobots tend to align and attach so that their magnetic dipole moments are parallel, reaching the minimum energy configuration as a microchain. Upon UV light illumination, phoretic interactions are restored, leading to microrobot clustering.

In addition, microrobot clusters usually manifested self-propulsion ability in $H_2O_2$ under UV light irradiation. Figure 5e is a time-lapse image showing the trajectory of a cluster (extracted from Supplementary Movie 5), which moved at an average speed of $3.4\,\mu m\,s^{-1}$, suggesting that clustering does not imply the loss of the motion feature.

A similar formation of microchains was previously observed for peanut-shaped $\alpha$-$Fe_2O_3$ microparticles under an applied magnetic field[56]. In a previously cited work, UV light-powered cubic $Pt/\alpha$-$Fe_2O_3$ microrobots were reported to have the ability of spontaneous assembly into microchains independent of UV light or $H_2O_2$ or magnetic fields[39]. In both cases, no reconfigurability or reversibility was possible. Instead, the results of the present study demonstrate an approach for formulating interaction-controlled, reconfigurable, reversible, and active self-assemblies of light-driven magnetic microrobots.

## Pesticide photocatalytic degradation

Light-powered microrobots are attractive for water purification since they simultaneously use light to move and degrade pollutants, allowing for a more efficient remediation process[57–59]. Among the emerging and most hazardous environmental pollutants are pesticides, whose utilization in agriculture has been progressively intensified to meet the ever-increasing demand for food[60,61]. Less than 0.1% of the applied pesticide reaches and destroys the pest, while the rest contaminates air, soil, and water[62]. Growing food using such polluted water allows pesticide propagation through the food chain, causing serious risks to human health even at trace levels[63]. Source-directed measures, such as taxes on pesticide utilization, and end-of-pipe measures, such as wastewater treatment, promote the transition toward sustainable agricultural practices[64]. Based on this, self-propelled $TiO_2/\alpha$-$Fe_2O_3$

microrobots were used to accelerate the photocatalytic degradation of 2,4D, an extensively used, persistent, and carcinogenic herbicide[65].

2,4D photocatalytic degradation by $TiO_2/\alpha$-$Fe_2O_3$ microrobots was studied under UV light irradiation in pure water for different durations (5, 10, 15, 30, and 60 min). Figure 6a shows the absorbance spectra of a 2,4D solution before (0 min) and after the treatments. 2,4D absorbance progressively decreases with time until being almost totally degraded within 30 min. Microrobots' ability to degrade 2,4D was quantified by calculating the degradation efficiency, plotted in Fig. 6b as a function of time. Microrobots degraded 97% of the pollutant within 30 min UV light irradiation.

Control experiments were performed to get more insights into the 2,4D degradation process. The obtained results are compared in Fig. 6c. The contribution of UV light irradiation to 2,4D degradation was determined by 60 min exposure in the absence of microrobots, revealing a negligible effect with 1% 2,4D degradation efficiency (absorbance spectrum in Supplementary Fig. 10a). Designing a control experiment that allows for identifying the contribution of active motion to 2,4D photocatalytic degradation is challenging. In fact, UV light irradiation activates both $TiO_2/\alpha$-$Fe_2O_3$ microrobots' spontaneous movement in pure water and photocatalytic reactions causing 2,4D degradation, simultaneously. Therefore, a control experiment in the dark allows for evaluating 2,4D adsorption by static microrobots only. In this condition, a 2,4D removal efficiency of 7% was found after 60 min in the dark (absorbance spectrum in Supplementary Fig. 10b). In contrast, under UV light exposure, $\alpha$-$Fe_2O_3$ microparticles are immobile yet photoactive. Using $\alpha$-$Fe_2O_3$ microparticles under UV light irradiation for 60 min, a 2,4D degradation efficiency of 55% was found (absorbance spectrum in Supplementary Fig. 10c). As a result, the superior performance of $TiO_2/\alpha$-$Fe_2O_3$ microrobots can be attributed to microrobots' self-propulsion ability and their increased photocatalytic activity after the deposition of $TiO_2$.

From the mechanism point of view, 2,4D degradation relies on the reaction between the irradiated photocatalyst and water, generating reactive oxygen species (ROS) to break the chemical bonds of the pollutant. To reveal the 2,4D degradation mechanism, radical trapping experiments were done by illuminating $TiO_2/\alpha$-$Fe_2O_3$ microrobots with UV light for 60 min in the presence of radical scavengers, i.e., chemical substances that can deactivate specific radicals. Specifically, EDTA, NBT, and isopropanol were utilized as scavengers for photogenerated holes ($h^+$), superoxide ions ($O_2^{-\cdot}$), and hydroxyl radicals ($OH^\cdot$), respectively[66,67]. Fig. 6b reports the 2,4D degradation efficiencies of radical trapping experiments, calculated from the absorbance spectra in Supplementary Fig. 10d. EDTA and NBT did not affect 2,4D degradation efficiency significantly (96% for EDTA, 94% for NBT). In contrast, 2,4D degradation efficiency dropped to 66% in the presence of isopropanol. Therefore, $OH^\cdot$ are identified as the main ROS involved in 2,4D degradation, which follows the mechanism proposed in Fig. 6d[68–70].

Compared to recently reported static photocatalysts, $TiO_2/\alpha$-$Fe_2O_3$ microrobots degraded the same amount of 2,4D in a shorter time due to their active motion, which allows them to face more pollutant molecules per unit of time[71–77]. While many studies on water purification by micro- and nanorobots still focus on easily degradable dyes, e.g., methylene blue, as models for water contaminants, or rely on using $H_2O_2$ to obtain the self-propulsion ability and boost the degradation efficiency, in the present study $TiO_2/\alpha$-$Fe_2O_3$ microrobots rapidly decomposed a persistent pollutant, i.e., 2,4D, without using $H_2O_2$[78]. In addition, the design of the microrobots is non-toxic, which is essential for practical applications. Conversely, $TiO_2$ coating by ALD is challenging to scale into mass production. Nonetheless, it could be achieved by cheaper methods, such as spin coating or sol-gel, to overcome this limitation[79,80].

In summary, the interaction-controlled, reconfigurable, reversible, and active self-assembly of light-powered magnetic $TiO_2/\alpha$-$Fe_2O_3$

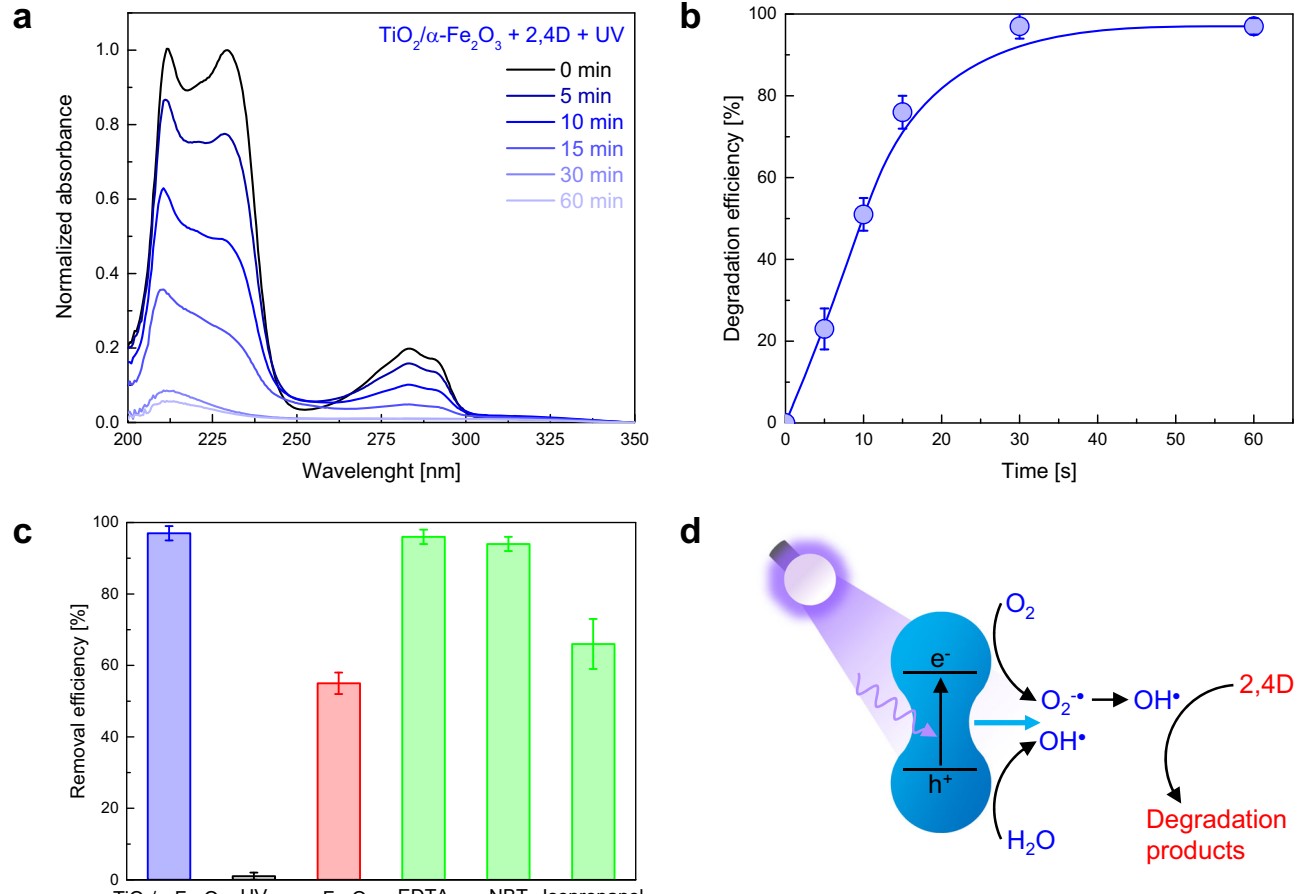

**Fig. 6 | Photocatalytic degradation of 2,4D by TiO₂/α-Fe₂O₃ microrobots.**
**a** Absorbance spectra of 2,4D solutions ($5 \times 10^{-5}$ M) before (0 min) and after the treatment with TiO₂/α-Fe₂O₃ microrobots under UV light irradiation in pure water for different durations (5, 10, 15, 30, and 60 min). **b** Degradation efficiency as a function of time. **c** Comparison of degradation efficiencies after different treatments: TiO₂/α-Fe₂O₃ microrobots (1 mg mL⁻¹) under UV light irradiation in pure water for 60 min (TiO₂/α-Fe₂O₃); UV light irradiation in pure water for 60 min (UV); α-Fe₂O₃ microparticles (1 mg mL⁻¹) and UV light irradiation in pure water for 60 min (α-Fe₂O₃); TiO₂/α-Fe₂O₃ microrobots (1 mg mL⁻¹), a radical scavenger (10 mg L⁻¹ EDTA, 10 mg L⁻¹ NBT, or 0.25 μL mL⁻¹ isopropanol), and UV light irradiation in pure water for 60 min (EDTA, NBT, and Isopropanol, respectively). Error bars represent the standard deviation, $n$ = 3 independent replicates. **d** 2,4D photocatalytic degradation mechanism by TiO₂/α-Fe₂O₃ microrobots under UV light irradiation in pure water.

microrobots was demonstrated. Photocatalytic microrobots were prepared by the scalable hydrothermal synthesis of highly uniform peanut-shaped α-Fe₂O₃ microparticles and their coating by a thin TiO₂ layer via ALD. After an initial deceleration, they showed constant light-driven self-propulsion in water due to a self-generated product gradient resulting from the catalyzed photochemical reactions, preferentially moving along the perpendicular direction to their long axis. Based on α-Fe₂O₃ ferromagnetism, microrobots' movement could be precisely steered using an external magnetic field, which, combined with their biocompatible design, makes them promising for biomedical applications. Under UV light irradiation in H₂O₂, microrobots spontaneously organized into large clusters owing to the attractive phoretic interactions related to H₂O₂ consumption on microrobots' surface. Moreover, these clusters manifested active motion under illumination, analogously to isolated microrobots. Removing the optical stimulus resulted in cluster disgregation and reconfiguration into snake-like microchains consisting of adjacent microrobots held by magnetic dipole–dipole interactions. Furthermore, they could switch from one self-organized state to the other multiple times following the activation or deactivation of the UV light source. These findings contribute to the development of more complex nature-mimicking metamachines. Besides microrobots' collective behaviors, their photoactivity makes them ideal swimming catalysts for water purification, as indicated by the efficient degradation of a highly persistent and

toxic pollutant, such as the herbicide 2,4D, in less than 30 min. Future efforts to deposit TiO₂ through less expensive techniques may allow the utilization of TiO₂/α-Fe₂O₃ microrobots in real-world settings.

## Methods

### Chemicals

Iron(III) chloride (FeCl₃·6H₂O, Alfa Aesar, ≥98%), sodium hydroxide (NaOH, Alfa Aesar, 98%), sodium sulfate (Na₂SO₄, Sigma Aldrich, ACS reagent, ≥99%), tetrakis(dimethylamino)titanium (TDMAT), hydrogen peroxide (H₂O₂, Sigma Aldrich, 30%), 2,4-Dichlorophenoxyacetic acid (2,4D, Sigma Aldrich, 97%), ethylenediaminetetraacetic acid (EDTA, ACS reagent, 99.4-100.6%), nitro blue tetrazolium chloride (NBT, Alfa Aesar, ≥98%), isopropanol (Sigma Aldrich, ≥99.5%).

### TiO₂/α-Fe₂O₃ microrobots fabrication

α-Fe₂O₃ peanut-shaped microparticles were synthesized as follows: 100 mL FeCl₃·6H₂O (2 M), 90 mL NaOH (6 M), and 10 mL Na₂SO₄ solutions (0.6 M) were prepared using ultra-pure water (18 MΩ cm) and mixed inside a 250 mL Pyrex bottle. The bottle was transferred into a preheated oven at 100 °C and maintained for 8 days[81]. The resulting reddish precipitate was collected by centrifugation (centrifugal force of 1006×g for 5 min), washed three times with ultra-pure water and absolute ethanol, and dried in an oven at 60 °C overnight.

An aqueous suspension of $\alpha$-Fe$_2$O$_3$ microparticles (1 mg mL$^{-1}$) was dropped on glass slides and dried overnight to fabricate the substrates for TiO$_2$ deposition. An ALD Ultratech/CambridgeNanoTech Fiji 200 reactor was used for this purpose. Argon and TDMAT (heated at 75 °C) were employed as the gas carrier and precursor, respectively. Oxygen was supplied through an inductively coupled plasma (20 s at 300 W power). At the beginning of the process, all heaters were set and stabilized at 250 °C for 1200 s. TDMAT and oxygen were introduced through ALD valves with a flow rate of 30 sccm and a pulse duration of 0.1 s, respectively. Pulse and purge times were kept constant for 600 cycles at a growth rate of ~0.052 nm per cycle to deposit ~30 nm TiO$_2$ layer. Finally, TiO$_2$/$\alpha$-Fe$_2$O$_3$ microrobots were released from the glass slides using a scalpel.

### Characterization techniques

$\alpha$-Fe$_2$O$_3$ microparticles and TiO$_2$/$\alpha$-Fe$_2$O$_3$ microrobots' morphology was characterized by a FEI Verios 460 L SEM. Before STEM analysis, samples were suspended in ultra-pure water, dropped on a holey carbon grid, and dried overnight. A TESCAN MIRA3 XMU SEM equipped with an Oxford Instruments EDX detector was used to examine TiO$_2$/$\alpha$-Fe$_2$O$_3$ microrobots' elemental composition. TiO$_2$/$\alpha$-Fe$_2$O$_3$ microrobots' crystalline structure was determined by XRD in grazing incidence mode (0.3° angle) using a Rigaku SmartLab 9 kW diffractometer equipped with a high-brightness Cu K$_\alpha$ rotating anode X-ray tube operated at 45 kV and 150 mA. $\alpha$-Fe$_2$O$_3$ microparticles and TiO$_2$/$\alpha$-Fe$_2$O$_3$ microrobots' surface chemical composition was investigated by XPS using a Kratos Analytical Axis Supra instrument with a monochromatic Al K$_\alpha$ (1486.7 eV) excitation source. All spectra were calibrated to the adventitious C 1$s$ peak at 284.8 eV and fitted using CasaXPS software. $\alpha$-Fe$_2$O$_3$ microparticles' magnetic hysteresis loop was measured using a Quantum Design VersaLab cryogen-free VSM at 300 K and an applied magnetic field ranging from −10 kOe to 10 kOe at steps of 10 Oe s$^{-1}$.

### Motion experiments

TiO$_2$/$\alpha$-Fe$_2$O$_3$ microrobots' light-driven motion was tested in pure water or H$_2$O$_2$ (0.5, 1, 2%) without any surfactant using a Nikon ECLIPSE Ti2 inverted optical microscope and a Hamamatsu C13440-20CU digital camera. A 365 nm UV light source or a 488 nm blue light source (Cool LED pE-300$^{lite}$ coupled to fluorescence filter cubes) at ~500 mW cm$^{-2}$ intensity was used to power microrobots' motion for different durations, up to 60 min. A control experiment was carried out under UV light irradiation in 0.1 M NaCl.

Magnetic field-controlled navigation experiments were conducted using a homemade magnetic setup consisting of three orthogonal coil pairs in a 3D-printed polylactic acid (PLA) backbone fitting a Nikon Ts2R inverted optical microscope equipped with a Basler acA1920-155uc camera. This apparatus generated a magnetic field **H** (3 mT) described by the following components

$$H_x = H_0 \, \text{sen}(\alpha)$$
$$H_y = H_0 \cos(\alpha) \qquad (1)$$

where $H_0$ is the magnetic field amplitude, proportional to the coils' current, and $\alpha$ is the navigation angle (0–360°), enabling magnetic field on the $xy$ plane. A 365 nm UV light source (Cool LED pE-100 coupled to a fluorescence filter cube) at ~1.5 W cm$^{-2}$ intensity was used to induce microrobots' motion. At the same time, the magnetic field allowed changing their motion direction through $\alpha$.

Movies of microrobots motion behavior were recorded at 10 fps and analyzed through NIS Elements Advanced Research and Fiji software to obtain their trajectories and calculate their speed.

### Numerical simulation

A numerical simulation was performed using the transport of diluted species module of COMSOL Multiphysics 5.5 software. TiO$_2$/$\alpha$-Fe$_2$O$_3$ microrobots were designed as peanut-like domains with dimensions of 2.44 and 1.12 µm, placed inside a $20 \times 20$ µm$^2$ rectangle. The simulation of H$_2$O$_2$ decomposition by a microrobot cluster under UV light irradiation was carried out by setting an H$_2$O$_2$ consumption rate of −0.18 mmol s$^{-1}$ m$^{-2}$ at the H$_2$O$_2$/microrobot boundaries and an H$_2$O$_2$ diffusion coefficient in water at 25 °C of $6.6 \times 10^{-10}$ m$^2$ s$^{-1}$. The H$_2$O$_2$ consumption rate was estimated according to the procedure described in Supplementary Discussion 1. It is worth mentioning that the obtained value represents an overestimation since it has been demonstrated that the H$_2$O$_2$ fuel-to-motion efficiency of micromotors is generally extremely low[82].

### Degradation experiments

A solution containing 2,4D ($5 \times 10^{-5}$ M) and TiO$_2$/$\alpha$-Fe$_2$O$_3$ microrobots (1 mg mL$^{-1}$) was prepared using ultra-pure water and transferred into UV-transparent cuvettes. The cuvettes were placed inside a customized irradiation chamber and exposed to the UV light emitted by three LZ4-04UV00 365 nm UV LEDs for different durations (5, 10, 15, 30, and 60 min). Afterward, the microrobots were separated from treated solutions by centrifugation (centrifugal force of 1006×$g$ for 5 min). The supernatants were collected, and their light absorption spectra were measured using a Jasco V-750 UV–visible spectrophotometer. An untreated 2,4D solution served as a reference. 2,4D degradation was assessed by monitoring the absorbance peak at 229.5 nm. Control experiments were performed in pure water without TiO$_2$/$\alpha$-Fe$_2$O$_3$ microrobots or UV light irradiation in pure water for 60 min, or by replacing the microrobots with static $\alpha$-Fe$_2$O$_3$ microparticles (1 mg mL$^{-1}$) under UV light irradiation in pure water for 60 min. Radical trapping experiments were carried out to elucidate the 2,4D degradation mechanism by exposing 2,4D and TiO$_2$/$\alpha$-Fe$_2$O$_3$ microrobots to UV light for 60 min in the presence of radical scavengers such as EDTA (10 mg L$^{-1}$), NBT (10 mg L$^{-1}$), and isopropanol (0.25 µL mL$^{-1}$). 2,4D degradation efficiency was calculated as

$$\text{Degradation efficiency} [\%] = (C_0 - C/C_0) \times 100 \qquad (2)$$

where $C_O$ and $C$ represent the initial concentration of 2,4D and the concentration at the time $t$ [min], respectively.

### Reporting summary

Further information on research design is available in the Nature Portfolio Reporting Summary linked to this article.

## Data availability

The data supporting the findings of the study are included in the article and the supplementary information files. The Source Data file has been deposited in Figshare under the accession code https://doi.org/10.6084/m9.figshare.24013836[83].

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

## Acknowledgements

The work was supported by the ERDF/ESF project TECHSCALE (No. CZ.02.01.01/00/22_008/0004587). This work was also financially supported by the European Union under the REFRESH - Research Excellence For Region Sustainability and High-tech Industries project number CZ.10.03.01/00/22_003/0000048 via the Operational Programme Just Transition. CzechNanoLab project LM2023051 funded by MEYS CR is gratefully acknowledged for the financial support of the measurements/sample fabrication at CEITEC Nano Research Infrastructure.

## Author contributions

M.Ur. synthesized α-$Fe_2O_3$ microparticles, characterized $TiO_2$/α-$Fe_2O_3$ microrobots, investigated their motion behavior, performed numerical simulations, interpreted data, and wrote the manuscript. M.Us. carried out $TiO_2$ ALD. X.P. conducted 2,4D degradation experiments. C.M.O. recorded SEM/STEM/EDX images and collected XPS spectra. M.Ur. and M.P. conceived the idea. M.P. supervised the research. All authors approved the final version of the manuscript.

## Competing interests

The authors declare no competing interests.
