## [Peer review file · Nature Communications]

REVIEWER COMMENTS

Reviewer #1 (Remarks to the Author):

The authors described a new TiO₂/α-Fe₂O₃ micromotor and reported its propulsion and assembly under light. They also explored the rapid degradation of the hazardous 2,4-Dichlorophenoxyacetic acid. Overall, the results of the paper are interesting and important. However, there are some major issues that must be fixed before it can be accepted by the journal.

1, Both photocatalytic and magnetic micromotors have been reported in the literature. The authors need to further explain the advantage of developing a system that has both functions. Specifically, they should consider whether there is an application that requires light-triggered motion and self-assembly of magnetic nanoparticles. By addressing this point, the authors can make a stronger case for the novelty and potential impact of their research.

2, In Figure 2b, the speed of the micromotor decreased rapidly in the first 5-10 seconds. Could the authors explain why this happened? Is it possible that the UV light penetrated the thin TiO₂ layer (~30 nm according to the method section) and caused the photo-corrosion of the inner Fe₂O₃, which released ions locally and reduced the motor speed? Control experiments under the same conditions using only Fe₂O₃ microparticles are encouraged to verify this phenomenon.

3, In addition to the above question, from Figure 2b, it seems that the motor speed is close to Brownian motion after 30-60 seconds. Did the authors test the lifetime of the micromotor? It is important to understand how long the motor can maintain its functionality and whether it can be used for practical applications. The authors should provide more details on this aspect of their work.

4, In figure 5c, the authors did not test the degradation efficiency using static micromotors. It would be more convincing to demonstrate the benefits of active motion of micromotors using static micromotors as a control experiment. This would help to verify whether the enhanced degradation efficiency is indeed due to the active motion of the micromotors, or simply due to the presence of TiO₂/Fe₂O₃ nanoparticles.

5, Which medium did the authors use for the 2,4-D degradation? If the authors used pure water, then they need to provide more characterization on the motor motion in pure water. If the authors used an H₂O₂ solution, then the mechanism illustration in Figure 4d should be revised because H₂O₂ is more unstable than H₂O and will be consumed first when the particles are illuminated with light.

6, the authors cited too many of their own works in the reference section, perhaps they can revise it.

Reviewer #2 (Remarks to the Author):

The manuscript entitled "Interaction-controlled reconfigurable reversible and active self-assembly of photocatalytic magnetic TiO₂/α-Fe₂O₃ microrobots for water purification" studies the moving and collective behavior of a peanut-shaped TiO₂/α-Fe₂O₃ microrobot under the irradiation of UV light. The authors also demonstrate the capability of the microrobots for water purification. The article is well-written and presented experiments are systematic and well designed. However, there is lack of novelty in the work. The propulsion mechanism for TiO₂-based light driven microrobots was previously and systematically studied [e.g Sachs J, Kottapalli S N, Fischer P, et al. Colloid and Polymer Science, 2021, 299, 269-280. and Maric T, Nasir M Z M, Webster R D, et al. Advanced Functional Materials, 2020, 30, 9, 1908614.], which is not a significant advance in the propulsion mechanism. Furthermore, there are numerous studies that demonstrate the magnetic navigated behavior of peanut-shaped microrobots [Lin Z, Fan X, Sun M, et al. ACS Nano, 2018, 12, 3, 2539-2545., Palacci J, Sacanna S,

Vatchinsky A, et al. *Journal of the American Chemical Society*, 2013, 135, 43, 15978-15981., and other works]. The major claim in the article is on the reconfigurable collective behavior of the microrobots. However, the cluster pattern [Mou F, Zhang J, Wu Z, et al. *Iscience*, 2019, 19, 415-424.] and the microchain pattern [Xie H, Sun M, Fan X, et al. *Science robotics*, 2019, 4, 28, eaav8006.] of TiO₂ or α -Fe₂O₃ microrobots were previously studied. The reconfigurable cluster control method proposed by the authors is realized through the combination of the two materials, which is feasible but not novel enough. Additionally, no new scientific principle is demonstrated, even the water purification principle is extensively used previously [Zhang Z, Zhao A, Wang F, et al. *Chemical Communications*, 2016, 52, 32, 5550-5553.]. I suggest that the authors find a different and more appropriate journal for publication of this work.

Reviewer #3 (Remarks to the Author):

The authors demonstrate the interaction-controlled, reconfigurable, reversible, and active self-assembly of TiO₂/ α -Fe₂O₃ microrobots which can also conduct light-powered self-propulsions and magnetic navigation. Then they show the photocatalytic water treatment application of the microrobots. The data is organized well, and the manuscript is well-written. However, several major issues should be addressed before being considered for publication.

(1) The authors claim that light-induced self-organization of micro- and nanorobots is still challenging. However, there are some exciting works on the light-induced self-organization of micro- and nanorobots (*Science* 2013; 339: 936-40; *Adv Mater* 2017; 29: 1701328; *Research* 2022; 2022: 9816562). Please clarify the new findings and main advantages of this work compared to the previous reports.

(2) The authors claim that the O₂ concentration gradient causes the propulsion of TiO₂/ α -Fe₂O₃ microrobots under light irradiation. However, the simulation of O₂ around the particle (Fig. 2d) shows a symmetric distribution along either the long axis or short axis. Please explain the mechanism more clearly. I do not think the concentration gradient between the long sides and the short sides can induce the motion shown in this manuscript. Maybe the particle has an asymmetric structure because of the asymmetric deposition of TiO₂?

(3) Does the H₂O₂ consumption contribute to the self-propulsion of TiO₂/ α -Fe₂O₃ microrobots?

(4) For numerical simulations, the authors should test the real values of the O₂ generation rate and H₂O₂ consumption rate.

(5) Is that possible for the authors to provide more clear and high-magnification microscopic images of TiO₂/ α -Fe₂O₃ microrobot clusters (Fig. 4e)? That could help to observe the cluster configuration.

(6) Can visible light drive the motor?

(7) The authors claimed that "Still, there was no control over microchains assembly/disassembly or reconfigurability (Line 80)." Actually, there are some published relevant works (*ACS Appl. Mater. Interfaces*. 2019, 11, 32937-32944; *Journal of Materials Chemistry C*. 2022, 10, 5079-5087.)

(8) The subscript of TiO₂/ α -Fe₂O₃ in Fig. 1a should be revised.

Response to the Reviewer #1

The authors described a new TiO₂/α-Fe₂O₃ micromotor and reported its propulsion and assembly under light. They also explored the rapid degradation of the hazardous 2,4-Dichlorophenoxyacetic acid. Overall, the results of the paper are interesting and important. However, there are some major issues that must be fixed before it can be accepted by the journal.

1, Both photocatalytic and magnetic micromotors have been reported in the literature. The authors need to further explain the advantage of developing a system that has both functions. Specifically, they should consider whether there is an application that requires light-triggered motion and self-assembly of magnetic nanoparticles. By addressing this point, the authors can make a stronger case for the novelty and potential impact of their research.

Authors reply: We agree with the Reviewer. Microrobots that possess photocatalytic and magnetic properties offer several advantages. They can utilize light to move and simultaneously degrade water pollutants by producing reactive oxygen species (ROS). Additionally, a magnetic field allows for easy collection of the microrobots after the treatment. Secondly, if the medium's characteristics obstruct the light-driven self-propulsion, magnetic fields can provide a robust motion while the light source triggers microrobots' photocatalytic activity. Moreover, magnetic fields enable precise navigation of microrobots towards hard-to-reach contaminated areas, such as inner parts of contaminated pipelines or specific regions of the human body where they can kill cancer cells via photocatalytic reactions.

To clarify these advantages, the following paragraph has been added to the introduction on page 3 of the revised manuscript:

“Thus, it can be used to devise microrobots with both photocatalytic and magnetic properties, which offer several benefits for various applications. They can utilize light to move and degrade water pollutants, while magnetic fields allow easy collection after the treatment. If the medium's properties hinder the light-driven self-propulsion, magnetic fields can provide powerful movement, and the light source can activate their photocatalytic activity. Furthermore, magnetic fields enable precise navigation of microrobots into hard-to-reach areas, such as inside pipelines in water remediation applications or specific regions of the human body in biomedical applications.”

In this study, microrobots' magnetic properties enabled the steering of their light-powered motion or formation of chains via magnetic dipole-dipole interactions, achieving reconfigurable self-assembly. This behavior can be utilized in water remediation, where the microrobots can be designed to self-assemble into a porous structure upon light exposure. This porous structure can allow water to flow through it, thereby preventing pollutants from escaping into the surrounding environment. Subsequently, the microrobots can degrade the entrapped pollutants into harmless byproducts via photocatalytic reactions. Their magnetic properties also allow for easy collection and removal. As another advantage, microrobots with reconfigurable self-assembly capabilities can be programmed to bind to solid impurities in water, like micro- and nanoplastics, by aggregating into complex structures around them. Once the microrobots have assembled around the plastic particles, light irradiation can activate them to lead to the photocatalytic degradation of the plastics. Alternatively, the microrobots can reconfigure into linear structures that can be actuated by magnetic fields, enabling them to remove plastic debris from the water. Furthermore, microrobots can be designed to detect and selectively bind to plastics in water, interfering with their self-assembly behavior and providing visual indicators of plastics' presence and distribution.

To illustrate potential applications of the reconfigurable self-assembly of photocatalytic magnetic microrobots and improve the novelty and impact of the work, the following paragraph has been added to the introduction on page 5 of the revised manuscript:

“Reconfigurable self-assembly of microrobots has significant potential for various applications, including water purification. By responding to external stimuli such as light or magnetic fields, microrobots can be programmed to assemble into porous networks or complex aggregates that trap water pollutants and degrade them through photocatalytic reactions. Additionally, microrobots can surround solid impurities, such as micro- and nanoplastics, and break them down when activated by light. Alternatively, microrobots can reconfigure into linear structures and use magnetic fields to mop up plastic debris from water sources. Moreover, microrobots can selectively detect and bind to these particles in water, providing a visual indication of their presence and distribution. Finally, microrobots’ magnetic properties facilitate their collection and removal from the treated water.”

2, In Figure 2b, the speed of the micromotor decreased rapidly in the first 5-10 seconds. Could the authors explain why this happened? Is it possible that the UV light penetrated the thin TiO₂ layer (~30 nm according to the method section) and caused the photo-corrosion of the inner Fe₂O₃, which released ions locally and reduced the motor speed? Control experiments under the same conditions using only Fe₂O₃ microparticles are encouraged to verify this phenomenon.

Authors reply: As suggested by the Reviewer, a control experiment was conducted using α -Fe₂O₃ microparticles under UV light irradiation in pure water. The trajectories of three microparticles after 40 s of illumination and the corresponding instantaneous speed as a function of time are presented in Fig. R1.

Fig. R1: a Micrograph showing the trajectories of three α -Fe₂O₃ microparticles under UV light irradiation in pure water for 40 s and b the corresponding instantaneous speed as a function of time. The scale bar is 5 μ m.

Random trajectories, characteristic of Brownian motion behavior, were observed for the microparticles. Additionally, a constant speed of $\sim 1 \mu\text{m s}^{-1}$ was measured. This control experiment demonstrated that UV light absorption alone is not sufficient to produce self-propulsion in bare α -Fe₂O₃ microparticles in pure water. Therefore, the active motion ability of TiO₂/ α -Fe₂O₃ microrobots is primarily attributed to the thin layer of TiO₂. At the same time, the observed speed decay in both pure water and H₂O₂ may be attributed to the TiO₂ corrosion. To test this hypothesis, a control experiment was conducted using the microrobots under UV light irradiation in 1% H₂O₂ for 5 min. Subsequently, the microrobots were collected, and their surfaces were characterized by XPS. Fig. R2 compares the Fe 2p and Ti 2p high-resolution XPS spectra of the microrobots before and after the experiment.

Fig. R2: High-resolution XPS spectra of **a** Fe $2p$ and **b** Ti $2p$ for $\text{TiO}_2/\alpha\text{-Fe}_2\text{O}_3$ microrobots before and after exposure to UV light irradiation in 1% H_2O_2 for 5 min.

As also shown in Supplementary Fig. 3 in the revised manuscript, the Ti $2p$ spectrum of the untreated microrobots exhibits Ti $2p_{3/2}$ and Ti $2p_{1/2}$ peaks, while no peaks are observed in the Fe $2p$ region, indicating a uniform deposition of the TiO_2 layer on $\alpha\text{-Fe}_2\text{O}_3$ microparticles' surface. In contrast, after 5 minutes of exposure to UV light irradiation in 1% H_2O_2 , Ti $2p_{3/2}$ and Ti $2p_{1/2}$ peaks are still present, as well as Fe $2p_{3/2}$ and Fe $2p_{1/2}$ peaks are found in the Fe $2p$ region. This result suggests that the TiO_2 coating underwent partial corrosion during the experiment, exposing a fraction of $\alpha\text{-Fe}_2\text{O}_3$ microparticles' surface. Therefore, it is reasonably concluded that the observed speed decrease is due to the partial degradation of the TiO_2 layer after exposure to UV light. It is worth noting that microrobots' self-propulsion remains constant over an extended period, as discussed in detail in response to the subsequent comment from the Reviewer.

Fig. R1 and Fig. R2 have been added to the revised Supplementary Information as Supplementary Fig. 5 and Supplementary Fig. 6, respectively. The following paragraph was added to the discussion on page 9 of the revised manuscript:

“A control experiment demonstrated that $\alpha\text{-Fe}_2\text{O}_3$ microparticles under prolonged UV light irradiation in pure water only display the characteristic random trajectories of Brownian motion with a low and constant speed of $\sim 1 \mu\text{m s}^{-1}$ (Supplementary Fig. 5). Consequently, $\alpha\text{-Fe}_2\text{O}_3$ may not have a major contribution in diminishing microrobots' speed. Instead, similar to the previously mentioned AgCl-based micromotor, a partial consumption of the engine, i.e., the TiO_2 coating, was assumed to explain the speed decrease of $\text{TiO}_2/\alpha\text{-Fe}_2\text{O}_3$ microrobots. To verify this hypothesis, an experiment was performed by exposing microrobots to UV light irradiation in 1% H_2O_2 for 5 min. Microrobots' surface was then characterized by XPS and compared to untreated microrobots in Supplementary Fig. 6. While microrobots originally showed Ti $2p_{3/2}$ and Ti $2p_{1/2}$ peaks in the Ti $2p$ region and no signal in the Fe $2p$ region, treated microrobots exhibited Fe $2p_{3/2}$ and Fe $2p_{1/2}$ peaks together with Ti $2p$ peaks. This result suggests that the uniform TiO_2 layer underwent partial corrosion, exposing a fraction of $\alpha\text{-Fe}_2\text{O}_3$ microparticles surface. Therefore, it is reasonably concluded that the initial deceleration is attributed to TiO_2 degradation.”

3, In addition to the above question, from Figure 2b, it seems that the motor speed is close to Brownian motion after 30-60 seconds. Did the authors test the lifetime of the micromotor? It is important to understand how long the motor can maintain its functionality and whether it can be used for practical applications. The authors should provide more details on this aspect of their work.

Authors reply: We agree with the Reviewer’s comment. Microrobots’ lifetime was studied by exposing them to UV light irradiation in pure water and recording movies at different durations (0, 5, 10, 15, 30, and 60 min), corresponding to those of 2,4D degradation experiments. Fig. R3 presents the resulting speed values.

Fig. R3: $\text{TiO}_2/\alpha\text{-Fe}_2\text{O}_3$ microrobots’ speed under UV light irradiation in pure water as a function of time. Error bars represent the standard deviation, $n = 20$ independent microrobots.

After the previously discussed speed decrease in response to UV light irradiation, microrobots speed is constant for at least 60 min. However, microrobots’ lifetime under UV light irradiation in 1% H_2O_2 was not investigated due to their self-assembly behavior, which makes speed calculations challenging over extended periods.

Fig. R3 has been added to the revised Supplementary Information as Supplementary Fig. 7. The following paragraph was added to the discussion on page 10 of the revised manuscript:

“Nevertheless, the lifetime of the microrobots was tested under UV light irradiation in pure water for 60 min, and it was found that microrobots maintained their self-propulsion ability and speed (Supplementary Fig. 7).”

4, In figure 5c, the authors did not test the degradation efficiency using static micromotors. It would be more convincing to demonstrate the benefits of active motion of micromotors using static micromotors as a control experiment. This would help to verify whether the enhanced degradation efficiency is indeed due to the active motion of the micromotors, or simply due to the presence of $\text{TiO}_2/\text{Fe}_2\text{O}_3$ nanoparticles.

Authors reply: We agree with the Reviewer’s comment. It has been extensively reported that the active motion of micro- and nanorobots has a positive impact on water purification applications, promoting contact with pollutants and their subsequent removal or degradation (references ⁵⁷⁻⁵⁹ in the revised manuscript). In this work, after 60 min under UV light irradiation, $\text{TiO}_2/\alpha\text{-Fe}_2\text{O}_3$ microrobots degraded 2,4D pesticide almost completely (97% degradation efficiency) thanks to their light-driven self-propulsion ability and photocatalytic activity. Since the microrobots can spontaneously move under UV light irradiation in pure water, i.e., in the absence of H_2O_2 as additional fuel, designing a reliable control experiment that allows for discriminating the contribution of the active motion from the photocatalytic performance was not possible. Indeed, stimulated by the Reviewer’s comment, a control experiment was carried out in

the absence of UV light for 60 min. In such a condition, TiO₂/ α -Fe₂O₃ microrobots can not move. At the same time, they can not induce 2,4D photocatalytic degradation. Therefore, this experiment allowed for evaluating microrobots absorption capability in the dark, only. Specifically, a minor decrease in the 2,4D solution absorbance peak intensity was observed, as shown in Fig. R4, resulting in a 2,4D removal efficiency of 7%.

Fig. R4: Absorbance spectra of 2,4D solutions (5×10^{-5} M) before (0 min) and after the treatment with **b** TiO₂/ α -Fe₂O₃ microrobots (1 mg mL^{-1}) without UV light irradiation for 60 min in pure water (“no motion condition”).

It is worth noting that this value was not reported in Fig. 5c in the revised manuscript since the figure shows solely 2,4D degradation efficiencies.

To discern the contribution of active motion and photocatalysis, a control experiment using Fe₂O₃ microparticles under UV light irradiation for 60 min was carried out, measuring a 2,4D degradation efficiency of 55%. In this experiment, particles could not move due to the absence of the TiO₂ layer, as discussed in the reply to the second comment from the Reviewer. Consequently, the decrease of 2,4D absorbance peak intensity is ascribed to static α -Fe₂O₃ microparticles-induced photocatalytic degradation. Nevertheless, it is worth noting that microrobots’ higher degradation efficiency (97 vs 55%) can be ascribed to both their self-propulsion and also to the photocatalytic activity of TiO₂, which is considered one of the most efficient photocatalytic semiconductors (reference ⁴³ in the revised manuscript).

Fig. R4 has been added to the revised Supplementary Information as Supplementary Fig. 10b. The methods section has been updated accordingly.

The following paragraph has been added to the discussion on page 16 of the revised manuscript.

“Designing a control experiment that allows for identifying the contribution of active motion to 2,4D photocatalytic degradation is challenging. In fact, UV light irradiation activates both TiO₂/ α -Fe₂O₃ microrobots’ spontaneous movement in pure water and photocatalytic reactions causing 2,4D degradation, simultaneously. Therefore, a control experiment in the dark allows for evaluating 2,4D adsorption by static microrobots only. In this condition, a 2,4D removal efficiency of 7% was found after 60 min in the dark (absorbance spectrum in Supplementary Fig. 10b). In contrast, under UV light exposure, α -Fe₂O₃ microparticles are immobile yet photoactive. Using α -Fe₂O₃ microparticles under UV light irradiation for 60 min a 2,4D degradation efficiency of 55% was found (absorbance spectrum in Supplementary Fig. 10c). As a result, the superior

performance of $\text{TiO}_2/\alpha\text{-Fe}_2\text{O}_3$ microrobots can be attributed to microrobots' self-propulsion ability and their increased photocatalytic activity after TiO_2 deposition.”

5, Which medium did the authors use for the 2,4-D degradation? If the authors used pure water, then they need to provide more characterization on the motor motion in pure water. If the authors used an H_2O_2 solution, then the mechanism illustration in Figure 4d should be revised because H_2O_2 is more unstable than H_2O and will be consumed first when the particles are illuminated with light.

Authors reply: 2,4D degradation was performed in pure water since microrobots demonstrated motion ability under UV light irradiation and also in the absence of H_2O_2 . Therefore, no H_2O_2 -related degradation mechanisms, such as Fenton and photo-Fenton reactions, were involved in 2,4D degradation. Consequently, to the best of our knowledge, the mechanism illustrated in Fig. 5d in the revised manuscript correctly describes the photocatalytic degradation of 2,4D in pure water.

Microrobots' UV light-powered motion has been studied in both pure water and H_2O_2 , and the obtained results are compared in Fig. 2 in the revised manuscript, reporting a microrobot's exemplificative trajectory and average instantaneous speed as a function of time in the two media. Moreover, Fig. 3 reports the trajectory of a microrobot in pure water under simultaneous exposure to UV light irradiation and an external magnetic field to steer its trajectory, while Supplementary Fig. 4 in the revised Supplementary Information shows the time-lapse images of a microrobot's trajectory in pure water for several on/off switching of the UV light source. Furthermore, additional experiments aiming to estimate the long-term motility of microrobots in the two media have been carried out following the third comment from the Reviewer.

Nevertheless, thanks to this comment, we understood that the figures' captions needed to be improved by better specifying the medium for microrobots' motion and application. Hence, the phrase “in pure water” was added in all figure captions related to microrobots motion experiments in pure water.

Besides, to highlight that no H_2O_2 was used during 2,4D degradation experiments, the following paragraph has been added to the discussion on page 17 of the revised manuscript:

“While many studies on water purification by micro- and nanorobots still focus on easily degradable dyes, e.g. methylene blue, as models for water contaminants, or rely on using H_2O_2 to obtain the self-propulsion ability and boost the degradation efficiency, $\text{TiO}_2/\alpha\text{-Fe}_2\text{O}_3$ microrobots rapidly decomposed a persistent pollutant, i.e., 2,4D, without using H_2O_2 .⁷⁸”

6, the authors cited too many of their own works in the reference section, perhaps they can revise it.

Authors reply: As suggested by the Reviewer, references have been carefully revised. In particular, six references to our previous works have been removed or replaced with new pertinent publications from other research groups, thereby the number of self-citations in the revised manuscript has been decreased to seven.

Response to the Reviewer #2

*The manuscript entitled “Interaction-controlled reconfigurable reversible and active self-assembly of photocatalytic magnetic TiO₂/α-Fe₂O₃ microrobots for water purification” studies the moving and collective behavior of a peanut-shaped TiO₂/α-Fe₂O₃ microrobot under the irradiation of UV light. The authors also demonstrate the capability of the microrobots for water purification. The article is well-written and presented experiments are systematic and well designed. However, there is lack of novelty in the work. The propulsion mechanism for TiO₂-based light driven microrobots was previously and systematically studied [e.g Sachs J, Kottapalli S N, Fischer P, et al. *Colloid and Polymer Science*, 2021, 299, 269-280. and Maric T, Nasir M Z M, Webster R D, et al. *Advanced Functional Materials*, 2020, 30, 9, 1908614.], which is not a significant advance in the propulsion mechanism.*

Authors reply: We appreciate the positive comment provided by the Reviewer regarding the quality of our manuscript. We understand the concern raised regarding the novelty of the work and would like to address it by comparing our study with the two references mentioned by the Reviewer.

In the study by Sachs et al. (*Colloid Polym. Sci.* **2021**, 299, 269), researchers focused on speed and displacement measurements of light-driven TiO₂/SiO₂ Janus particles in dense suspensions using heterodyne laser Doppler velocimetry. These particles demonstrate motion by self-phoresis resulting from the asymmetric generation of a product gradient due to photo-induced H₂O₂ decomposition at the TiO₂-coated side. Therefore, they do not show self-propulsion in the absence of H₂O₂ fuel. Moreover, SiO₂ only serves as a spherical support for fabricating Janus particles upon TiO₂ deposition.

Similarly, in the study by Maric et al. (*Adv. Func. Mater.* **2020**, 30, 1908614), researchers developed light-powered micromotors based on TiO₂ microparticles with different metal coatings (Pt, Au, and Ag) to determine the optimal metal choice for enhancing micromotors' speed. Micromotors' motion under UV light exposure is based on self-electrophoresis: different photochemical reactions at the metal and semiconductor sides of the micromotors produce an asymmetric distribution of charged products and a local electric field driving micromotors' self-propulsion in pure water.

In contrast to Sachs et al., TiO₂/α-Fe₂O₃ microrobots consist of two active components: 1) TiO₂ coating with remarkable photocatalytic activity and 2) α-Fe₂O₃ support with photocatalytic and magnetic properties. This combination enables self-propulsion and allows for steering microrobots' motion direction using an external magnetic field. Importantly, they can move also in pure water without using expensive noble metals, such as in the study by Maric et al. Additionally, compared to these studies, versatile motion features of microrobots were explored in detail, such as speed decay over time and their preferential orientation to the motion direction.

By highlighting these distinctions, we believe this work offers a novel contribution to the field of light-driven microrobots for water purification. For this reason, the following paragraph was added to the introduction on page 3 of the revised manuscript:

“For instance, TiO₂, one of the most used photocatalysts due to its high photocatalytic efficiency, stability, and safety, has been combined with different noble metals to produce efficient UV light-powered micro- and nanorobots.^{22–25} Alternatively, TiO₂ has been deposited onto passive particles to make them move under UV light irradiation.^{26”}

The study by Sachs et al. (*Colloid Polym. Sci.* **2021**, 299, 269) was added to the revised manuscript reference list as reference ²⁶:

26. Sachs, J., Kottapalli, S. N., Fischer, P., Botin, D. & Palberg, T. Characterization of active matter in dense suspensions with heterodyne laser Doppler velocimetry. *Colloid Polym. Sci.* **299**, 269–280 (2021).

Furthermore, there are numerous studies that demonstrate the magnetic navigated behavior of peanut-shaped microrobots [Lin Z, Fan X, Sun M, et al. *ACS Nano*, 2018, 12, 3, 2539-2545., Palacci J, Sacanna S, Vatchinsky A, et al. *Journal of the American Chemical Society*, 2013, 135, 43, 15978-15981., and other works].

Authors reply: We have carefully reviewed the two studies mentioned by the Reviewer. The first study by Lin et al. (*ACS Nano* **2018**, 12, 2539) focused on the use of rotating magnetic fields to actuate peanut-shaped α -Fe₂O₃ micromotors for cargo transport. In contrast to the TiO₂/ α -Fe₂O₃ microrobots, their research did not involve light or investigate micromotors' photocatalytic properties.

The second study by Palacci et al. (*J. Am. Chem. Soc.* **2013**, 135, 15978) demonstrated the magnetic steering of peanut-shaped α -Fe₂O₃ microparticles powered by blue light, which is similar to our experiment. However, their light-driven self-propulsion ability required several additional steps, including etching the microparticles' surface with a concentrated acid solution (5 M HCl), using 1% H₂O₂, and adjusting the pH to 8.5 with 5 mM tetramethylammonium hydroxide (TMAH). In contrast, TiO₂/ α -Fe₂O₃ microrobots move in pure water without the need for pH adjustments.

We appreciate the Reviewer for bringing these studies to our attention. We believe this work contributes to the field by demonstrating a simpler and more efficient method for achieving light-powered self-propulsion ability and magnetic steering in peanut-shaped α -Fe₂O₃ microrobots without the requirement of chemical treatments pre-utilization or medium pH adjustments.

Based on this, the following paragraphs have been added to the discussion on pages 10 and 11 of the revised manuscript, respectively:

“Earlier studies on α -Fe₂O₃ micromotors required surface activation through the use of concentrated acid solutions, high amounts of H₂O₂ (ranging from 1% to 10%), and an increase of the medium pH (~8.5) to achieve the self-propulsion capability.^{48,50} However, the application of a TiO₂ coating circumvents the need for potentially hazardous pre-utilization steps, toxic H₂O₂, and pH adjustments.”

“Previous studies have utilized magnetic fields to guide peanut-shaped α -Fe₂O₃ micromotors for various applications, including the non-contact manipulation of cells.⁵¹ In the case of TiO₂/ α -Fe₂O₃ microrobots, their combined light-powered motion and magnetic responsiveness provide additional possibilities for accurate navigation within a liquid medium.”

The study by Lin et al (*ACS Nano* **2018**, 12, 2539) has been included in the revised manuscript reference list as reference ⁵¹, while the study by Palacci et al (*J. Am. Chem. Soc.* **2013**, 135, 15978) was already included in the submitted manuscript reference list:

51. Lin, Z. et al. Magnetically Actuated Peanut Colloid Motors for Cell Manipulation and Patterning. *ACS Nano* **12**, 2539–2545 (2018).

The major claim in the article is on the reconfigurable collective behavior of the microrobots. However, the cluster pattern [Mou F, Zhang J, Wu Z, et al. iScience, 2019, 19, 415-424.] and the microchain pattern [Xie H, Sun M, Fan X, et al. Science robotics, 2019, 4, 28, eaav8006.] of TiO₂ or α -Fe₂O₃ microrobots were previously studied.

Authors reply: In the study by Mou et al. (*iScience* **2019**, 19, 415), TiO₂ micromotors form clusters that expand upon UV light irradiation in the presence of 0.5% H₂O₂, resulting in collective motion. To arrange themselves in a linear structure, these clusters must pass through a narrow channel. In contrast, TiO₂/ α -Fe₂O₃ microrobots reported in our study can spontaneously and reversibly reconfigure into two distinct configurations: clusters and chains. Unlike the previous study, the chain configuration in this work involves a single line of adjacent microrobots and is not influenced by the geometry of the surrounding environment, such as a microchannel.

Xie et al. (*Science Robotics* **2019**, 4, eaav8006) reported on α -Fe₂O₃ microrobots that form swarms capable of quickly and reversibly reconfiguring into chain, vortex, and ribbon-like structures using an alternating magnetic field. Magnetic fields offer various parameters that can be adjusted to control the assembly of magnetic microrobots. However, they require bulky and expensive magnetic setups. In contrast, light is an attractive and easily accessible energy source for powering micro- and nanorobots. Nevertheless, achieving precise control over microrobot assemblies using only light is challenging.

In this study, we focused on the reconfigurable collective behavior of TiO₂/ α -Fe₂O₃ microrobots, highlighting their ability to be manipulated using light irradiation, which governs the interactions (phoretic or magnetic dipole-dipole interactions) between microrobots, leading to the formation of cluster and chain-like configurations. By switching the light source on and off, it is possible to reversibly transit between these two configurations.

On these bases, the following paragraphs have been added to the introduction on page 4 of the revised manuscript:

“Magnetic fields offer several adjustable parameters that govern the assembly of magnetic micro- and nanorobots. However, their implementation necessitates relatively bulky and expensive magnetic setups. Achieving such a degree of control solely using light sources is desirable yet challenging.”

“Interestingly, when enriched with hydroxyl groups (OH), TiO₂ micromotors gathered into flocks, which dilated upon UV light irradiation in 0.5% H₂O₂ leading to micromotors’ collective motion.³³ Also, these micromotors arranged themselves into elongated shapes when transiting a narrow microfluidic channel.”

Both studies indicated by the Reviewer were already included in the submitted manuscript reference list.

The reconfigurable cluster control method proposed by the authors is realized through the combination of the two materials, which is feasible but not novel enough. Additionally, no new scientific principle is demonstrated, even the water purification principle is extensively used previously [Zhang Z, Zhao A, Wang F, et al. Chemical Communications, 2016, 52, 32, 5550-5553.]. I suggest that the authors find a different and more appropriate journal for publication of this work.

Authors reply: In this study, TiO₂/α-Fe₂O₃ microrobots were employed to degrade the herbicide 2,4D in water under UV light irradiation. To the best of our knowledge, this specific application of microrobots has not been explored before. While previous studies, including the one mentioned by the Reviewer (*Chem. Commun.* **2016**, 52, 5550), have extensively used the water purification principle, they often targeted dyes that are relatively easier to degrade, such as methylene blue (MB).

Additionally, in most of the previous studies, microrobots' motion required the presence of H₂O₂, which contributes to the degradation process by generating hydroxyl radicals (OH[•]) under UV light irradiation. This work, on the other hand, demonstrated the ability to completely degrade the persistent herbicide 2,4D in less than 30 minutes using TiO₂/α-Fe₂O₃ microrobots solely under UV light irradiation in the absence of H₂O₂.

Based on these factors, we believe that this study represents a significant advancement in the field of water purification by micro- and nanorobots and that this topic matches well with the scope and objectives of *Nature Communications*. We hope that the aforementioned clarifications address the concerns of the Reviewer and highlight the unique contribution of this study.

The following paragraph has been added to the discussion on page 17 of the revised manuscript.

“While many studies on water purification by micro- and nanorobots still focus on easily degradable dyes, e.g. methylene blue, as models for water contaminants, or rely on using H₂O₂ to obtain the self-propulsion ability and boost the degradation efficiency, TiO₂/α-Fe₂O₃ microrobots rapidly decomposed a persistent pollutant, i.e., 2,4D, without using H₂O₂.⁷⁸”

The study by Zhang et al (*Chem. Commun.* **2016**, 52, 5550) has been added to the revised manuscript reference list as reference ⁷⁸:

78. Zhang, Z., Zhao, A., Wang, F., Ren, J. & Qu, X. Design of a plasmonic micromotor for enhanced photo-remediation of polluted anaerobic stagnant waters. *Chem. Commun.* **52**, 5550–5553 (2016).

Response to the Reviewer #3

The authors demonstrate the interaction-controlled, reconfigurable, reversible, and active self-assembly of TiO₂/α-Fe₂O₃ microrobots which can also conduct light-powered self-propulsions and magnetic navigation. Then they show the photocatalytic water treatment application of the microrobots. The data is organized well, and the manuscript is well-written. However, several major issues should be addressed before being considered for publication.

*(1) The authors claim that light-induced self-organization of micro- and nanorobots is still challenging. However, there are some exciting works on the light-induced self-organization of micro- and nanorobots (*Science* 2013; 339: 936-40; *Adv Mater* 2017; 29: 1701328; *Research* 2022; 2022: 9816562). Please clarify the new findings and main advantages of this work compared to the previous reports.*

Authors reply: We thank the Reviewer for this comment, which allowed us to improve the comparison of our work with previous literature. Palacci et al. conducted a pioneering study (*Science* **2013**, 339, 936-940) where they introduced colloidal “surfers” composed of polymer/α-Fe₂O₃ microparticles exhibiting self-assembly, breaking, and re-arrangement into two-dimensional “living” crystals when exposed to light. This behavior was attributed to the interplay between self-propulsion and attractive osmotic and phoretic interactions. The experiments involved immersing the particles in a medium containing H₂O₂ (0.1-3%) as a fuel, 5 mM tetramethylammonium hydroxide (TMAH) as a pH modifier to reach a pH value of ~8.5, and 3.4 mM sodium dodecyl sulfate (SDS) as the surfactant.

Another interesting study by Singh et al. (*Advanced Materials* **2017**, 29, 1701328) demonstrated dynamic crystals formed by the attractive interaction between active TiO₂/SiO₂ Janus microparticles, which acted as nucleation centers, and passive colloids under UV light irradiation in the presence of 1.5% H₂O₂ and 1 mM TMAH to get a neutral pH value. The authors also demonstrated the ability to modify the crystal’s size and symmetry by varying the light intensity and the size ratio between active and passive particles.

Moreover, Che et al. conducted a recently published and fascinating study (*Research* **2022**, 2022, 9816562) that focused on light-programmable micromotor assemblies in water. The authors achieved this behavior by utilizing TiO₂ microspheres uniformly coated with Pt nanoparticles. The micromotors displayed the ability to assemble into static or motile planar crystals, as well as phototactic streams. The attractive or repulsive forces acting on the particles were dictated by the incident angle of UV light illumination.

In the present work, unique findings regarding TiO₂/α-Fe₂O₃ microrobots were presented. Specifically, the microrobots show self-propulsion in pure water without requiring pH adjustment or surfactants, while in H₂O₂ they spontaneously assemble into active planar structures (clusters) that can reversibly switch to static linear structures (chains) by toggling the UV light source. Notably, unlike the previously mentioned studies, the microrobots do not disperse when the light source is switched off. Instead, they reconfigure into a different structure. This behavior is attributed to the microrobots’ magnetic properties, which enable them to interact via magnetic dipole-dipole interactions even in the absence of light when attractive phoretic interactions cease. Furthermore, magnetic properties offer opportunities for precise magnetic navigation and more complex self-organization behaviors in combination with light sources, which have yet to be explored thoroughly.

To expand the description of previous literature, the following paragraph has been added to the introduction on page 4 of the revised manuscript:

“Colloidal “surfers” based on polymer/ α -Fe₂O₃ microparticles self-organized into two-dimensional (2D) “living” crystals when exposed to light in 0.1-3% H₂O₂ at basic pH conditions.³⁴ Whereas, under UV light irradiation in 1.5% H₂O₂ and neutral pH conditions, active TiO₂/SiO₂ Janus particles attracted passive colloids, functioning as nucleation centers for the growth of 2D crystals whose size and symmetry were controlled by light intensity and size ratio between active and passive particles.³⁵ Another approach exploited the incident light angle for programming the self-assembly of Pt/TiO₂ micromotors into active or static planar crystals and phototactic micromotor streams.³⁶ The formation of these superstructures relies on the interplay between light-induced self-propulsion and attractive osmotic or phoretic interactions. Therefore, by turning off the light source, the self-assembled configurations are broken.

The studies by Palacci et al (*Science* **2013**, 339, 936-940), Singh et al (*Advanced Materials* **2017**, 29, 1701328), and Che et al (*Research* **2022**, 2022, 9816562) have been added to the revised manuscript reference list as references ³⁴, ³⁵, and ³⁶, respectively:

34. Palacci, J., Sacanna, S., Steinberg, A. P., Pine, D. J. & Chaikin, P. M. Living crystals of light-activated colloidal surfers. *Science* **339**, 936–940 (2013).
35. Singh, D. P., Choudhury, U., Fischer, P. & Mark, A. G. Non-Equilibrium Assembly of Light-Activated Colloidal Mixtures. *Adv. Mater.* **29**, 1701328 (2017).
36. Che, S. *et al.* Light-Programmable Assemblies of Isotropic Micromotors. *Research* **2022**, (2022).

(2) *The authors claim that the O₂ concentration gradient causes the propulsion of TiO₂/ α -Fe₂O₃ microrobots under light irradiation. However, the simulation of O₂ around the particle (Fig. 2d) shows a symmetric distribution along either the long axis or short axis. Please explain the mechanism more clearly. I do not think the concentration gradient between the long sides and the short sides can induce the motion shown in this manuscript. Maybe the particle has an asymmetric structure because of the asymmetric deposition of TiO₂?*

Authors reply: We agree with the Reviewer’s comment. As highlighted by the Reviewer, UV light-induced product concentration gradient between peanuts’ long and short axes may not be sufficient to drive their self-propulsion. On the other hand, we believe that the hypothesis regarding the asymmetric deposition of TiO₂, leading to an asymmetric spatial distribution of reaction products, may not be valid in our case. Indeed, SEM-EDX analyses did not indicate an asymmetric TiO₂ coating on α -Fe₂O₃ microparticles’ surface, unlike metal/semiconductor Janus micro- and nanorobots fabricated by metal sputtering deposition on semiconductor micro- and nanoparticles, since TiO₂ was deposited by atomic layer deposition (ALD). In agreement with previous reports (including reference ⁴¹ in the revised manuscript), a TiO₂ layer obtained by atomic layer deposition (ALD) should result in a uniform coating of α -Fe₂O₃ microparticles. By excluding the possibility of a Janus structure, it is worth noting that in the experimental setup, UV light is shined from the bottom of the microscope glass slides where microrobots lie, i.e., through the objective of an inverted microscope. Therefore, it is reasonable to assume that asymmetric illumination is the mechanism behind TiO₂/ α -Fe₂O₃ active motion, which results from a higher concentration of reaction products (O₂) in the bottom, irradiated area of the microrobots. Moreover, the presence of a solid wall (microscope glass slide in motion experiments or bottom of the vessel in 2,4D degradation experiments) probably influences microrobots’ movement, leading to a “sliding” behavior which has been previously investigated for catalytic motors moving on planar walls by Uspal et al (*Soft Matter* **2015**, 11, 434).

On these bases, the numerical simulation of the O₂ concentration gradient around a microrobot in Fig. 2d in the submitted manuscript was modified to take into account the light irradiation from the bottom, the presence of a solid wall or substrate (microscope glass slide), and a more realistic value for O₂ generation rate, according to the fourth comment from Reviewer #3. The revised Fig. 2d is reported below for convenience. The methods section of the revised manuscript has been updated accordingly.

Fig. 1: d Simulated O₂ concentration gradient around a TiO₂/α-Fe₂O₃ microrobot under UV light irradiation in 1% H₂O₂.

To comment on the updated Fig. 2d, the following paragraph was added to the discussion on page 10 of the revised manuscript:

“It is worth noting that during motion experiments, microrobots are irradiated from the bottom of the microscope glass slides. Therefore, to simulate the O₂ concentration gradient generated by a microrobot, an asymmetrically UV light-irradiated microrobot on a planar surface, representing the microscope glass slide, was utilized, as shown in Fig. 2d. Along the illuminated surface of the microrobot, the O₂ concentration is higher, resulting in the observed “sliding” movement of microrobots on the substrate.⁴⁹”

The study by Uspal et al ((*Soft Matter* **2015**, *11*, 434) has been added to the revised manuscript reference list:

49. Uspal, W. E., Popescu, M. N., Dietrich, S. & Tasinkevych, M. Self-propulsion of a catalytically active particle near a planar wall: From reflection to sliding and hovering. *Soft Matter* **11**, 434–438 (2015).

(3) Does the H₂O₂ consumption contribute to the self-propulsion of TiO₂/α-Fe₂O₃ microrobots?

Authors reply: In principle, H₂O₂ decomposition under UV light irradiation generates a larger product gradient according to the following chemical reactions involving photogenerated electron-hole (e⁻-h⁺) pairs:

While the larger product concentration gradient due to H₂O₂ usually results in a higher micro- and nanorobots’ speed compared to pure water, in the present study such a difference is appreciable only in the first seconds of UV light irradiation. Fig. 2b in the revised manuscript

shows that the average instantaneous speed of microrobots in 1% H₂O₂ is initially higher than in pure water (~15 vs 7.5 μm s⁻¹). After that, the speed decreases in both cases till reaching a similar value of 2.5-3 μm s⁻¹. However, the speed decay is faster in 1% H₂O₂ than in pure water, due to more rapid corrosion of the photoactive TiO₂ layer, according to the reply to the second comment from Reviewer #1.

It is worth noting that, in 1% H₂O₂, microrobots rapidly tend to form clusters upon shining UV light irradiation. Therefore, the measurement of microrobots' speed in such conditions was more challenging and required recording and analyzing only those microrobots which were not "trapped" by clusters and "free" to move for a sufficiently long time.

To highlight the contribution of H₂O₂ to microrobots' self-propulsion, the following paragraph was added to the discussion on page 10 of the revised manuscript:

"If H₂O₂ is present, it can contribute to the expansion of the O₂ concentration gradient through supplementary chemical reactions, as shown in Fig. 2c. This leads to increased speed and faster deceleration of microrobots compared to pure water during the initial stage of UV light irradiation."

(4) For numerical simulations, the authors should test the real values of the O₂ generation rate and H₂O₂ consumption rate.

Authors reply: The Reviewer's suggestion to test the real values of the O₂ generation rate and H₂O₂ consumption rate in numerical simulations has been addressed. To measure the H₂O₂ consumption rate, a 1 mL solution containing 1 mg mL⁻¹ microrobots and 1% H₂O₂ was introduced into UV transparent cuvettes and exposed to UV light irradiation for different periods (0, 10, 30, and 60 min). After the treatment, the microrobots were separated from the solutions by centrifugation. The solutions were analyzed by a UV-Visible spectrophotometer to acquire their absorbance spectra and determine the H₂O₂ concentration following a previous work by Aye et al (*Anal. Chem.* **2005**, 77, 5814). Fig. R5 shows the time dependence of H₂O₂ concentration.

Fig. R5: H₂O₂ concentration as a function of time following H₂O₂ consumption experiments by TiO₂/α-Fe₂O₃ microrobots under UV light irradiation in 1% H₂O₂.

After 30 min of UV light exposure, H₂O₂ is almost completely consumed. By calculating the slope of the linear fit in the range of 0÷30 min, an H₂O₂ consumption rate of -0.011 mol L⁻¹ min⁻¹

¹ was obtained, corresponding to $1.8 \times 10^{-7} \text{ mol s}^{-1}$ for a 1 mL solution. To obtain the rate normalized per unit area [$\text{mol s}^{-1} \text{ m}^{-2}$], as requested by the software for numerical simulations, the microrobots' surface area was estimated. By neglecting the contribution of the thin TiO_2 layer and assuming the peanut shape formed by two adjacent spheres with a radius of $0.56 \mu\text{m}$ (measured by STEM analysis) the volume occupied by a peanut was calculated to be $\sim 1.5 \times 10^{-18} \text{ m}^3$. Using an $\alpha\text{-Fe}_2\text{O}_3$ density of 5.3 g cm^{-3} , the mass of a single microrobot was found as $\sim 8.0 \times 10^{-12} \text{ g}$. Based on these calculations, 1 mg of microrobots should contain $\sim 1.3 \times 10^8$ microrobots. The surface area of a microrobot, given by the sum of the surface area of the two adjacent spheres with a radius of $0.56 \mu\text{m}$, was found to be $\sim 7.9 \times 10^{-12} \text{ m}^2$, resulting in a microrobots surface area of $\sim 1.0 \times 10^{-3} \text{ m}^2$, in total. Therefore, the surface area-normalized H_2O_2 consumption rate was calculated to be $-1.8 \times 10^{-4} \text{ mol s}^{-1} \text{ m}^{-2}$. Since the dissociation of each H_2O_2 molecule results in an O_2 molecule ($\text{H}_2\text{O}_2 + 2\text{h}^+ \rightarrow 2\text{H}^+ + \text{O}_2$), the O_2 generation rate is equal and opposite to the H_2O_2 consumption rate, i.e., $+1.8 \times 10^{-4} \text{ mol s}^{-1} \text{ m}^{-2}$. The numerical simulations in Fig. 2d and 4d in the submitted manuscript have been performed using these more realistic values, which were reported below and in the revised manuscript as Fig. 3d and 5d, respectively. The methods section of the manuscript has been updated accordingly.

Fig. 2: d Simulated O_2 concentration gradient around a $\text{TiO}_2/\alpha\text{-Fe}_2\text{O}_3$ microrobot under UV light irradiation in 1% H_2O_2 .

Fig. 3: d Simulated H_2O_2 concentration gradient around a cluster of $\text{TiO}_2/\alpha\text{-Fe}_2\text{O}_3$ microrobots under UV light irradiation in 1% H_2O_2 , resulting in a pressure imbalance inducing the attraction of a nearby $\text{TiO}_2/\alpha\text{-Fe}_2\text{O}_3$ microrobot.

Fig. R5 has been added to the revised Supplementary Information as the Supplementary Fig. 11. The following paragraph was added to the revised Supplementary Information as Supplementary Discussion 1:

“ H_2O_2 consumption experiments were performed to measure the H_2O_2 consumption rate by $\text{TiO}_2/\alpha\text{-Fe}_2\text{O}_3$ microrobots under UV light irradiation. For this purpose, UV-transparent cuvettes were filled with 1 mL of solution containing 1 mg mL^{-1} microrobots and 1% H_2O_2 , and exposed

to UV light irradiation for different durations (0, 10, 30, and 60 min). Afterward, microrobots were separated by centrifugation to record the UV-Vis absorbance spectra of treated solutions. The H₂O₂ concentration was determined from absorbance spectra according to a previous work.⁴ Supplementary Fig. 11 shows the time dependence of the H₂O₂ concentration. The slope of the linear fit in the range 0–30 min gave an H₂O₂ consumption rate of 0.011 mol L⁻¹ min⁻¹, corresponding to 1.8×10⁻⁷ mol s⁻¹ for 1 mL of solution. To normalize the rate per unit area [mol s⁻¹ m⁻²], TiO₂/α-Fe₂O₃ microrobots' surface area was estimated. For simplicity, the microrobots were assumed to be as peanut-shaped α-Fe₂O₃ microparticles consisting of two adjacent spheres with a radius of 0.56 μm, as measured by STEM analysis. Considering that the volume (*V*) of a spherical particle with radius *r* is $V = \frac{4}{3} \pi r^3$, the volume occupied by a microrobot was found as ~1.5×10⁻¹⁸ m³. Given an α-Fe₂O₃ density (*ρ*) of 5.3 g cm⁻³, the mass (*m*) of the single microrobot was calculated to be 8.0×10⁻¹² g through the relation $m = \rho V$. Therefore, in 1 mL of solution with 1 mg mL⁻¹ microrobots, there are 1.3×10⁸ microrobots. Since the surface (*A*) of a spherical particle with radius *r* is $A = 4 \pi r^2$, the area exposed by a microrobot was calculated to be 7.9×10⁻¹² m², whereas the surface area of 1 mg of microrobots was found as 1.0×10⁻³ m². Consequently, the surface area-normalized H₂O₂ consumption rate was calculated to be approximately -1.8×10⁻⁴ mol s⁻¹ m⁻². At the same time, since the dissociation of an H₂O₂ molecule produces an O₂ molecule (H₂O₂ + 2h⁺ → 2H⁺ + O₂), the O₂ generation rate was estimated to be +1.8×10⁻⁴ mol s⁻¹ m⁻².”

The study by Aye et al (*Anal. Chem.* **2005**, *77*, 5814) has been added to the revised Supplementary References as reference ⁴:

4. Aye, T. T., Low, T. Y. & Sze, S. K. Nanosecond laser-induced photochemical oxidation method for protein surface mapping with mass spectrometry. *Anal. Chem.* **77**, 5814–5822 (2005).

(5) *Is that possible for the authors to provide more clear and high-magnification microscopic images of TiO₂/α-Fe₂O₃ microrobot clusters (Fig. 4e)? That could help to observe the cluster configuration.*

Authors reply: In response to the Reviewer's request, time-lapse micrographs showing the clustering process of TiO₂/α-Fe₂O₃ microrobots under UV light irradiation in 1% H₂O₂ have been recorded and reported in Fig. R6, as shown below. These images were captured at the clearest and highest possible magnification according to the experimental setup. We believe that these additional images will provide a more clear view of the cluster configuration. Fig. R6 has been added to the revised Supplementary Information as Supplementary Fig. 9.

Fig. R6: Time-lapse micrographs showing the clustering process of $\text{TiO}_2/\alpha\text{-Fe}_2\text{O}_3$ microrobots under UV light irradiation in 1% H_2O_2 . Scale bars are 10 μm .

The following paragraph has been added to the discussion on page 13 of the revised manuscript:

“Supplementary Fig. 9 includes time-lapse micrographs that demonstrate the process of microrobots aggregation and cluster configuration.”

(6) *Can visible light drive the motor?*

Authors reply: Stimulated by this comment, the capability of visible light to drive the self-propulsion of $\text{TiO}_2/\alpha\text{-Fe}_2\text{O}_3$ microrobots was investigated using blue light at varying H_2O_2 concentrations. When exposed to blue light in pure water, the microrobots did not exhibit a noticeable autonomous movement. However, upon the introduction of a low amount of H_2O_2 , the microrobots demonstrated a self-propelled motion. It is important to note that TiO_2 can not be activated by blue light due to its large bandgap, thus the observed motion under visible light is attributed to $\alpha\text{-Fe}_2\text{O}_3$, which has a smaller bandgap suitable for blue light absorption. The trajectory of a characteristic microrobot under blue light irradiation in 0.5% H_2O_2 is reported in Fig. R7a.

Fig. R7: **a** Micrograph showing the trajectory of a $\text{TiO}_2/\alpha\text{-Fe}_2\text{O}_3$ microrobot under blue light irradiation in 0.5% H_2O_2 for 44 s and **b** the corresponding instantaneous speed as a function of time. The scale bar is 5 μm . **c** Microrobots' speed as a function of H_2O_2 concentration under blue light irradiation. Error bars represent the standard deviation, $n = 20$ independent microrobots.

Interestingly, unlike UV light, exposure to blue light did not cause a decay in microrobot speed (Fig. R7b). To gain further insight into the motion behavior under blue light, microrobots' speed

was measured at different H₂O₂ concentrations ranging from 0.5 to 2%, finding that a higher amount of H₂O₂ results in a slight increase in microrobots' speed (Fig. R7c).

Fig. R7 has been added to the revised Supplementary Information as Supplementary Fig. 8. The methods section of the revised manuscript has been updated accordingly.

To describe the visible-light-driven self-propulsion ability of microrobots, the following paragraph was added to the discussion on page 10 of the revised manuscript:

“Microrobots' mobility was also examined under visible light irradiation, using blue light (Supplementary Fig. 8). No self-propulsion was observed in pure water; however, upon the introduction of H₂O₂, microrobots demonstrated the ability to self-propel. This behavior can be attributed to the absorption of blue light by α -Fe₂O₃, considering the larger bandgap of TiO₂. Notably, no speed decay was detected in the initial stage of the experiment, unlike for UV light irradiation. Furthermore, microrobots' speed was measured at increasing concentrations of H₂O₂, finding a minor increase in speed from 0.5 to 2% H₂O₂.”

(7) *The authors claimed that “Still, there was no control over microchains assembly/disassembly or reconfigurability (Line 80).” Actually, there are some published relevant works (ACS Appl. Mater. Interfaces. 2019, 11, 32937-32944; Journal of Materials Chemistry C. 2022, 10, 5079-5087.)*

Authors reply: We apologize because the sentence structure can lead to a misunderstanding of the readers. Indeed, the sentence “*Still, there was no control over microchains assembly/disassembly or reconfigurability.*” in the submitted manuscript refers to the previous paragraph about Pt/ α -Fe₂O₃ cubic microrobots self-assembly into microchains. However, stimulated by the Reviewer's comment, the discussion was improved by considering the two suggested studies.

In both cases, the micromotors consist of a combination of magnetic and photocatalytic materials that allow for chain formation and motion. Specifically, Wang et al (*ACS Applied Materials & Interfaces* **2019**, *11*, 32937-32944) reported TiO₂ microspheres coated by thin layers of Ni and Au, while Guo et al (*Journal of Materials Chemistry C* **2022**, *10*, 5079-5087) deposited a TiO₂ layer onto magnetic microbeads. Then, the micromotors' assembly into chains was driven by an applied magnetic field, while UV light irradiation let them move in water, eventually with the assistance of H₂O₂.

On one hand, in these two examples, the assembly/disassembly of chains is possible by simply turning on/off the magnetic field. On the other hand, the self-assembly process is not spontaneous, as it is for TiO₂/ α -Fe₂O₃ microrobots. Moreover, there is no reconfigurability in the mentioned studies, as the micromotor collectives can only switch between the assembled (chain) and disassembled states. In contrast, our microrobots can transform reversibly from a planar configuration to a linear one, demonstrating reconfigurability.

On these bases, the following paragraphs were added to the introduction on pages 4 and 5 of the revised manuscript:

“Besides, using a combination of magnetic and photocatalytic materials, micromotors were arranged into chain-like structures through the application of a magnetic field, while their movement in water or H₂O₂ was facilitated by light irradiation.^{37,38} In this manner, the assembly or disassembly of microchains is achieved by activating or deactivating the magnetic field.”

“Still, for spontaneously formed microchains, there was no control over their assembly/disassembly, unlike those manipulated by magnetic fields, or reconfigurability.”

The studies by Wang et al (*ACS Applied Materials & Interfaces* **2019**, *11*, 32937-32944) and Guo et al (*Journal of Materials Chemistry C* **2022**, *10*, 5079-5087) have been added to the revised manuscript reference list as references ³⁷ and ³⁸, respectively:

37. Wang, L., Kaeppler, A., Fischer, D. & Simmchen, J. Photocatalytic TiO₂ Micromotors for Removal of Microplastics and Suspended Matter. *ACS Appl. Mater. Interfaces* **11**, 32937–32944 (2019).

38. Guo, X. *et al.* Phototactic micromotor assemblies in dynamic line formations for wide-range micromanipulations. *J. Mater. Chem. C* **10**, 5079–5087 (2022).

(8) *The subscript of TiO₂/α-Fe₂O₃ in Fig. 1a should be revised.*

Authors reply: We thank the Reviewer for bringing this typo to our attention. The subscript in Fig. 1a has been corrected in the revised manuscript.

REVIEWER COMMENTS

Reviewer #1 (Remarks to the Author):

The authors have clearly addressed my concerns, I have no further questions about this manuscript now.

Reviewer #2 (Remarks to the Author):

We are very grateful to the author for the revision of the manuscript and the responses given. The author responded in detail to our comments, and the revised article has been greatly improved. We think that this manuscript is now appropriate for publication by Nature Communications, since this manuscript reported a novel TiO₂/α-Fe₂O₃ microrobot, which can self-propel and navigate in H₂O₂ and pure water. The authors showed their collective behavior and water treatment ability in detail. This article has potential to promote the future swarm control and environmental remediation applications of the micro/nanorobots.

Reviewer #3 (Remarks to the Author):

The authors have addressed some of my concerns. However, the major issue of the propulsion mechanism has not been addressed.

(1) The propulsion mechanism of the motor is still unclear. If the O₂ distribution was generated as that shown in Figure 2d, the motor would be in a "hovering" state instead of a "sliding" state near the substrate surface, as reported by the reference paper provided by the author in Reply 2 (Soft Matter 2015, 11, 434).

(2) The consumption of H₂O₂ did not represent the generation of O₂. The O₂ concentration variation during the photocatalytic reaction in the medium can be monitored by a dissolved-oxygen meter.

(3) As confirmed by the authors in the "Pesticide photocatalytic degradation" section, the hydroxyl radicals (\cdot OH) were produced by the micromotors under light irradiation. If so, the products would be different from those given in Figure 2c, and some charged species may be produced in the photocatalytic reactions. Thus, the motor is probably propelled through electrolyte diffusiophoresis.

(4) As the work mentioned by the authors (Ref. 41), the deposition of TiO₂ by the atomic layer deposition method still results in asymmetric distribution of the TiO₂ layer because of the physical contact between neighboring Fe₂O₃ particles and that between the particle and the substrate.

REPLY TO REVIEWERS

In the following pages (one section per Reviewer), the Reviewer report is reported in *italics*, interrupted by indented insertions with our responses (**Author reply**) and relative **changes** (in red), if any, done throughout the manuscript at the point of the page number indicated. We hope this response format is clear enough.

Response to the Reviewer #1

The authors have clearly addressed my concerns, I have no further questions about this manuscript now.

Authors reply: We thank the Reviewer for this comment and for recommending the article for publication in Nature Communications.

Response to the Reviewer #2

We are very grateful to the author for the revision of the manuscript and the responses given. The author responded in detail to our comments, and the revised article has been greatly improved. We think that this manuscript is now appropriate for publication by Nature Communications, since this manuscript reported a novel TiO₂/α-Fe₂O₃ microrobot, which can self-propel and navigate in H₂O₂ and pure water. The authors showed their collective behavior and water treatment ability in detail. This article has potential to promote the future swarm control and environmental remediation applications of the micro/nanorobots.

Authors reply: We thank the Reviewer for this comment and for recommending the article for publication in Nature Communications.

Response to the Reviewer #3

The authors have addressed some of my concerns. However, the major issue of the propulsion mechanism has not been addressed.

(1) The propulsion mechanism of the motor is still unclear. If the O₂ distribution was generated as that shown in Figure 2d, the motor would be in a “hovering” state instead of a “sliding” state near the substrate surface, as reported by the reference paper provided by the author in Reply 2 (Soft Matter 2015, 11, 434).

Authors reply: We agree with the Reviewer’s comment. The original claim about “microrobots “sliding” movement on the substrate surface” and the numerical simulation in Fig. 2d have been removed in the revised manuscript. Based on the third comment of the Reviewer, a new explanation for the propulsion mechanism of TiO₂/α-Fe₂O₃ microrobots has been provided.

(2) The consumption of H₂O₂ did not represent the generation of O₂. The O₂ concentration variation during the photocatalytic reaction in the medium can be monitored by a dissolved-oxygen meter.

Authors reply: Based on the first and third Reviewer’s comments and the newly proposed propulsion mechanism, the numerical simulation of the O₂ generation by an irradiated TiO₂/α-Fe₂O₃ microrobot in Fig. 2d has been removed from the revised manuscript.

We completely agree with the Reviewer. We acknowledge that the consumption of H₂O₂ does not necessarily equate to the generation of O₂. In our study, we were assuming that the decomposition of H₂O₂ predominantly contributes to O₂ production to estimate an approximate value for the O₂ generation rate. While we recognize that this estimation may not be as precise as the value obtained using a dissolved-oxygen meter, it is worth noting that our primary objective with the numerical simulation was only to illustrate the spatial distribution of the O₂ gradient around an irradiated TiO₂/α-Fe₂O₃ microrobot. In addition, it is worth mentioning that the fuel (H₂O₂)-to-motion efficiency of the micromotors has been generally to be low compared to combustion motors (Wang et al, *J. Am. Chem. Soc.* **2013**, 135(28), 10557–10565).

We sincerely appreciate the Reviewer’s valuable comment. The following paragraph has been added on page 21 of the revised manuscript:

“It is worth mentioning that the obtained value represents an overestimation since it has been demonstrated that the H₂O₂ fuel-to-motion efficiency of micromotors is generally extremely low.⁸²”

Reference ⁸² has been added to the revised manuscript reference list.

(3) As confirmed by the authors in the “Pesticide photocatalytic degradation” section, the hydroxyl radicals (·OH) were produced by the micromotors under light irradiation. If so, the products would be different from those given in Figure 2c, and some charged species may be produced in the photocatalytic reactions. Thus, the motor is probably propelled through electrolyte diffusiophoresis.

Authors reply: We thank the Reviewer for this important suggestion, which led us to further investigate and identify TiO₂/α-Fe₂O₃ microrobots propulsion mechanism as electrolyte self-diffusiophoresis. To verify this hypothesis, microrobots’ light-powered motion was examined in a concentrated salt solution (0.1 M NaCl). The high ionic strength of the media obstructed their

propulsion ability, confirming the proposed mechanism. On these bases, the chemical reactions in Fig. 2c of the revised manuscript have been modified as reported below.

Fig. 2: c Scheme of $\text{TiO}_2/\alpha\text{-Fe}_2\text{O}_3$ microrobots light-powered motion mechanism.

The discussion about the $\text{TiO}_2/\alpha\text{-Fe}_2\text{O}_3$ microrobots' light-powered motion mechanism on page 10 of the revised manuscript has been modified as follows:

“In light of this, the movement of $\text{TiO}_2/\alpha\text{-Fe}_2\text{O}_3$ microrobots is explained according to the scheme in Fig 2c. Upon UV light irradiation, the TiO_2 layer absorbs photons with sufficient energy to excite electrons from the semiconductor's valence band to the conduction band. The photogenerated electron-hole pairs react with surrounding water molecules, producing a charged product concentration gradient responsible for microrobots' autonomous motion via electrolyte self-diffusiophoresis.⁴⁸ A control experiment in a concentrated salt solution (0.1 M NaCl) confirmed the proposed mechanism since the high ionic strength of the media hindered microrobots' motility. The H_2O_2 fuel, when present, contributes to the propulsion process through supplementary chemical reactions, as shown in Fig. 2c.”

Reference ⁴⁸, which supports the chemical reactions shown in Fig. 2c, was added to the revised manuscript reference list.

⁴⁸ Rajagopal, S., Paramasivam, B. & Muniyasamy, K. Photocatalytic removal of cationic and anionic dyes in the textile wastewater by H_2O_2 assisted TiO_2 and micro-cellulose composites. *Sep. Purif. Technol.* **252**, 117444 (2020).

(4) As the work mentioned by the authors (Ref. 41), the deposition of TiO_2 by the atomic layer deposition method still results in asymmetric distribution of the TiO_2 layer because of the physical contact between neighboring Fe_2O_3 particles and that between the particle and the substrate.

Authors reply: We appreciate the Reviewer's comment. According to ref. 41, we expected to observe uncovered regions on the surface of $\alpha\text{-Fe}_2\text{O}_3$ microparticles through SEM analysis after TiO_2 deposition by ALD. However, due to the low TiO_2 film thickness and high surface roughness of $\alpha\text{-Fe}_2\text{O}_3$ microparticles, compared to the flat surface of the Mg microspheres used in ref. 41, these uncovered regions were challenging to visualize.

Nevertheless, we can not exclude the possibility that TiO_2 -free areas are present in $\text{TiO}_2/\alpha\text{-Fe}_2\text{O}_3$ microrobots, located in correspondence to the contact regions between $\alpha\text{-Fe}_2\text{O}_3$ microparticles and the substrate during TiO_2 ALD, as suggested by the Reviewer. Therefore, the paragraph on page 6 of the revised manuscript was modified as follows:

“Although such uncovered areas were not directly observed in $\text{TiO}_2/\alpha\text{-Fe}_2\text{O}_3$ microrobots SEM images, their presence is expected because of the physical contact between $\alpha\text{-Fe}_2\text{O}_3$ microparticles and the substrate, leading to a shadowing effect during TiO_2 ALD deposition.”

On the other hand, we prefer not to modify the schematic illustrations of $\text{TiO}_2/\alpha\text{-Fe}_2\text{O}_3$ microrobots.

REVIEWERS' COMMENTS

Reviewer #3 (Remarks to the Author):

The authors have addressed my concerns.

Response to the Reviewer #3

The authors have addressed my concerns.

Authors reply: We thank the Reviewer for this comment and for recommending the article for publication in Nature Communications.